# A multi-year short-range hindcast experiment with CESM1 for evaluating climate model moist processes from diurnal to interannual time scales

Hsi-Yen Ma[1], Chen Zhou[2], Yunyan Zhang[1], Stephen A. Klein[1], Mark D. Zelinka[1], Xue Zheng[1], Shaocheng Xie[1], Wei-Ting Chen[3], and Chien-Ming Wu[3]

[1]Lawrence Livermore National Laboratory, Livermore, California, USA
[2]School of Atmospheric Sciences, Nanjing University, Nanjing, China
[3]Department of Atmospheric Sciences, National Taiwan University, Taipei, Taiwan

*Correspondence to*: Hsi-Yen Ma (ma21@llnl.gov)

**Abstract.** We present a multi-year short-range hindcast experiment and its experimental design for better evaluation of both the mean state and variability of atmospheric moist processes in climate models from diurnal to interannual time scales and facilitate model development. We use the Community Earth System Model version 1 as the base model and performed a suite of 3-day hindcasts initialized every day starting at 00Z from 1997 to 2012. Three processes – the diurnal cycle of clouds during different cloud regimes over the Central U.S., precipitation and diabatic heating associated with the Madden-Julian Oscillation (MJO), and the response of precipitation, surface radiative and heat fluxes, as well as zonal wind stress to sea surface temperature anomalies associated with the El Niño-Southern Oscillation – are evaluated as examples to demonstrate how one can better utilize simulations from this experiment to gain insights into model errors and their connection to physical parameterizations or large-scale state. This is achieved by comparing the hindcasts with corresponding long-term observations for periods based on different phenomena. These analyses can only be done through this multi-year hindcast approach to establish robust statistics of the processes under well-controlled large-scale environment because these phenomena are either interannual climate variability or only happen a few times in a given year (e.g. MJO, or cloud regime types). Furthermore, comparison of hindcasts to the typical simulations in climate mode with the same model allows one to infer what portion of a model's climate error directly comes from fast errors in the parameterizations of moist processes. As demonstrated here, model biases in the mean state and variability associated with parameterized moist processes usually develop within a few days, and manifest within weeks to affect the simulations of large-scale circulation and ultimately the climate mean state and variability. Therefore, model developers can achieve additional useful understanding of the underlying problems in model physics by conducting a multi-year hindcast experiment.

# 1 Introduction

The representation of moist processes - clouds, convection, precipitation and the associated radiative perturbations - and their interactions with the large-scale circulation in Global Climate Models (GCMs) or Earth System Models (ESMs) remains one of the grand challenges for the modeling community (Bony et al. 2015). Aside from using high resolution cloud modeling (cloud-resolving models or large-eddy simulations) to study detailed cloud dynamics and physics and their interactions with large-scale environment, significant progress on process-level understanding of moist processes has also been achieved recently through the application of a climate model hindcast approach (Phillips et al. 2004; Williams et al. 2013) to gain insights relevant to the improvement of parameterizations in climate models. This progress has been made either through individual modeling studies (e.g., Xie et al. 2004, 2008; Klein et al. 2006; Barton et al. 2012, 2014; Medeiros et al. 2012; Hannah and Maloney 2014; Chandra et al. 2015; Van Weverberg et al. 2015; Zheng et al. 2016; 2017; Qin et al. 2018; Chen et al. 2019; Zhang et al. 2020), or through coordinated model intercomparison projects (Lin et al. 2012; Williams et al. 2013; Ma et al. 2014; Klingaman et al. 2015; Xavier et al. 2015; Morcrette et al. 2018; Ma et al. 2018; Van Weverberg et al. 2018).

Earlier studies with climate model hindcast experiments usually focused on a relatively short period of time, such as those spanning Intensive Observation Periods or field campaigns. However, determining the robust aspects of certain cloud processes may not be achieved through these short simulations. Recent work in climate model studies has demonstrated the benefit of using multiple years of short-range hindcasts. For example, O'Brien et al. (2016) presented a direct comparison between observed and simulated weather events across multiple resolutions through the analysis of 5-day long hindcasts performed every day during a five-year period. This hindcast modeling framework allows them to assess the degree to which increased resolution improves the fidelity of extreme events in the model. Further, Phillips et al. (2017) studied the land-atmosphere coupling over the U.S. Southern Great Plains (SGP) using a suite of 16 years of short-range hindcasts as well as a free-running Atmospheric Model Intercomparison Project (AMIP, Gates 1992) simulation for the same period in comparison with the U.S. Department of Energy Atmospheric Radiation Measurement (ARM) observations. Although the surface climate state of the hindcasts deviates less from the observations in contrast to the AMIP simulation, they further identify that the model surface characteristics (e.g., vegetation cover) or physical parameterizations involving land-atmosphere coupling are more important factors than the performance of surface climate state in controlling the coupling behaviors. Chen et al. (2019) assessed precipitation biases in the Community Earth System Model version 1 (CESM1) during the abrupt onset of the South China Sea summer monsoon, a key precursor of the overall East Asia summer monsoon. A multi-year hindcast approach was utilized to obtain the well-constrained synoptic-scale horizontal circulation each year during the onset period. Their results highlighted the need for an appropriate representation of land-ocean-convection interactions over coastal areas in order to improve the simulation of monsoon onset.

The above examples indicate the benefit of using multiple years of short-range hindcasts for robust process-level modeling studies in comparison with long-term observations. Also, evaluating ensemble short-range hindcasts with the same climate model can complement the traditional way of conducting AMIP-type model evaluation. In the present paper, we present a multi-year short-range hindcast experiment and its experimental design for better evaluating both the mean state and variability of atmospheric moist processes in climate models to facilitate model development using the CESM1 as the base model. This experiment provides an new opportunity to address several modeling issues associated with moist processes, which cannot be achieved from previous short Transpose AMIP II hindcasts (Williams et al. 2013), or one or two years of short-range hindcasts that we conducted in the past (Xie et al. 2012, Ma et al, 2013, Ma et al. 2015). This is because these phenomena are either interannual climate variability or only happen a few times in a given year, and thus we need multi-years to robustly quantify the errors associated with these phenomena. We will demonstrate the unique value of diagnosing systematic model errors from diurnal to interannual time scales with this suite of multi-year short-range hindcasts paired with long-term observations, such as from various satellites or from major field programs like the U.S. Department of Energy ARM Program. Process-level understanding can be achieved by comparing hindcasts with observations for periods based on the phenomena of interest rather than the climatological mean state. Three processes – the diurnal cycle of clouds during different cloud regimes at the ARM SGP Site, precipitation and diabatic heating associated with the Madden-Julian Oscillation (MJO), and moist processes response to sea surface temperature (SST) anomalies associated with the El Niño-Southern Oscillation (ENSO) – are evaluated as examples to gain insights into model errors and their connection to physical parameterizations. We also demonstrate that systematic errors in the mean state of moist processes over the global scale are very robust and do not show significant interannual variations in either error magnitudes or patterns over large spatial domains. Our focus of this manuscript is to document this multi-year experiment as a model description paper and provide examples on how to better utilize this suite of hindcasts. The remainder of this manuscript is organized into three sections. Section 2 describes the hindcast experiment design, experiments performed and validation datasets. Section 3 presents three examples of how we can better utilize this suite of multi-year short-range hindcasts to evaluate the variability of moist processes over various time scales. Section 4 presents a summary.

## 2 Model experiments and validation data

### 2.1 Model and experiment design

All simulations were conducted with the CESM1 using the active atmospheric and land model components (version cesm1_0_5, FC5 compset, Neale et al. 2012). The atmospheric model component is the CAM5 with the finite volume dynamical core at a horizontal resolution of 0.9° latitude by 1.25° longitude and 30 vertical levels. The land model component is the Community Land Model version 4.0 (CLM4) with the same horizontal resolution. The ocean and sea ice components are prescribed with the National Oceanic and Atmospheric Administration (NOAA) Optimum Interpolation weekly SSTs and sea ice (Reynolds et al. 2002).

The hindcast procedure is based on Ma et al. (2015). In summary, we applied the horizontal velocities, temperature, specific humidity and surface pressure from the European Centre for Medium-Range Weather Forecasts (ECMWF) ERA-Interim Reanalysis (Dee et al. 2011) for the initial atmospheric states. We applied bilinear interpolation for the horizontal remapping for all the state variables to the model grid. For vertical remapping, we follow the procedure used at ECMWF when initializing model with foreign analysis: Quadratic interpolation is used for temperature, linear interpolation is used for specific humidity, and a combination of linear and quadratic interpolation is used for zonal and meridional winds. To avoid spurious gravity waves associated with differences in topography between ERA-Interim and CAM5, we applied a spatial smoothing for the state variables (Gerrity and McPherson 1970). We also adjusted the surface pressure associated with differences in topography between ERA-Interim and CAM5 using the hydrostatic approximation. A nudging simulation (horizontal velocities nudging only following Zhang et al. 2014) with CAM5/CLM4 was also performed to acquire other necessary variables (e.g., cloud and aerosol fields), which are not available from the ERA-Interim Reanalysis for the atmospheric initial conditions. The nudging simulation started from January 1$^{st}$, 1996 and stopped at December 31$^{st}$, 2012 with a 6 h relaxation time scale. During the nudging simulation, the reanalysis data are linearly interpolated between two time steps to match the model's current time.

We do not use land-surface conditions from the nudging simulation for the land-surface initial condition in the hindcasts. This is because in a nudging simulation, biased precipitation, winds, and surface fluxes are allowed to pass to the land model, which will cause larger biases in the simulated soil moisture and temperature (Ma et al. 2015). Instead, land initial conditions are taken from an offline land model simulation (I_2000 compset) forced by reanalysis and observations including precipitation, surface winds, and surface radiative fluxes (CRUNCEP, N. Viovy 2013, unpublished data) rather than coupled it to an active atmospheric model. The default bilinear interpolation method is used to interpolate the forcing datasets to the CLM grid. The offline land model simulation started from 1990 to 2012 and we performed five cycles (1990 to 2012) for the offline simulation to allow proper spin-up of the land conditions. The carbon and nitrogen cycles in this particular CLM4 setup are not active. After that, we continued the offline land model simulation to the desire starting date and use the land model restart file (.r file) as the land initial condition. We have published a documentation and relevant scripts in generating initial conditions on GitHub (https://github.com/PCMDI/CAPT), which includes the initialization generation procedure, the nudging procedure, and land-surface spinup procedure. The way to conduct a single hindcast is the same as performing an AMIP simulation except we use the initial conditions from the procedure described above. The multi-year hindcast experiment is a suite of 3-day long hindcasts starting at 00Z every day for the years of 1997 to 2012 (Figure 1) using the initial conditions obtained from the procedure described above. We concatenated each hindcast from 24-48 (48-72) hours lead time to form a pseudo Day 2 (Day 3) time series of 16-year duration from 1997 to 2012. Day 1 data are not analyzed to minimize the impact of model spin-up (Ma et al. 2013; 2014).

We also conducted a 16-year long AMIP simulation with the same model for the same period. In this AMIP simulation, the state of the atmosphere evolves freely without constraints. Note that the nudging simulation mentioned above has the same model configuration as the AMIP simulation with the exception of the nudging terms. Also, the greenhouse gas and solar forcing is based on the setup of the CESM1 FC5 compset, which corresponds to the year 2000 level for all the simulation period. This is because the CMIP5 forcing data does not go beyond 2005. We also want to exclude the impact of the interannual variations in the solar and greenhouse gas forcings to our simulations so that we can better identify possible causes of model biases associated with parameterizations. To compare with high-temporal frequency observations collected at the ARM permanent sites as well as at various major field campaign locations within the simulation period, we have additionally generated output for every model timestep (30 minutes interval) in additional to output for the entire global domain. Figure 2 and Table 1 identify their geographical locations and output grids. The location to obtain all the model output and necessary initial conditions to conduct the multi-year hindcasts are described in the Code and data availability Section.

## 2.2 Strategy on performing the multi-year hindcasts on a high performance computing system

Since each hindcast is independent and can be completed very fast from less than half an hour to two hours depending on the speed of the high performance computing (HPC) system, one can easily bundle as many hindcasts as possible into one job submission on a HPC system. This is the concept of submitting multiple parallel jobs simultaneously in a single batch script. Most HPC systems nowadays have this capability and even encourage people to submit large jobs with discount on the charge of computer hours. For example, a single hindcast takes one hour to complete with 5 computer nodes. One can then request 1,825 nodes for performing short-range hindcasts for one year period in a single job submission. The queue time may require a longer wait but the entire hindcasts will finish within an hour. One can submit multiple jobs in the queue for multiple years of hindcasts. One issue to keep in mind is the storage space for model output. There is usually a disk quota for scratch space on a HPC system. Therefore, a script or code to save model output to a long-term storage system, such as a High Performance Storage System (HPSS), may be necessary to prevent reaching the scratch disk quota while the model is running.

## 2.3 Comparison datasets

Daily global observational precipitation is adopted from the Global Precipitation Climatology Project Version 1.2 (GPCP, Adler et al. 2003). Absorbed shortwave flux at top of atmosphere (SWAbs), outgoing longwave radiation (OLR) as well as net surface shortwave and longwave fluxes are obtained from Clouds and the Earth's Radiant Energy System (CERES) Energy Balanced And Filled (EBAF) observations (Loeb et al. 2009; Kato et al. 2013, Edition 2.8). Total cloud fraction is from the International Satellite Cloud Climatology Project (ISCCP) D2 dataset (Rossow et al. 1999). Global winds and surface turbulent heat fluxes are from the ERA-Interim Reanalysis. Vertical profiles of cloud fraction at the ARM SGP site are from the ARM Best Estimate (ARMBE; Xie et al. 2010) Active Remote Sensing of Clouds data (ARSCL; Clothiaux et al. 2000, 2001). The available time period of the dataset is listed in Table 2. We interpolated all the datasets onto model's grid for better comparison.

## 3 Example analysis

Our goal here is to demonstrate the usefulness of the multi-year hindcasts in providing a different perspective on several long-standing moist processes errors in GCMs through three examples. Note that identifying causes for individual model issue requires further investigation and is beyond the current scope of this model experiment description paper. For sensitivity tests to various parameter choices for a specific scheme or parameterization, it is not necessary to perform this suite of multi-year hindcast experiment once the issue has been identified. Instead, one could perform a "core experiment" (i.e., series of short-range hindcast over one-year period) as we proposed in Ma et al. (2015), or perform a set of hindcasts just for the set of key dates with the phenomena of interest (e.g. days with sallow cumulus at ARM SGP, or phase 3 of various MJOs, which we will introduce in the following text).

### 3.1 Cloud regimes at the ARM SGP site

One common application of hindcasts for model evaluation is during major field campaigns where intensive observations are available at very high temporal scale. However, field campaigns are usually confined to a certain short period and cannot determine the robust aspects of certain cloud processes, which are available only from long-term monitoring as provided by satellites or permanent ground-based sites. From over ten years of cloud radar observations at the ARM SGP site, Zhang and Klein (2010) computed the diurnal cycle of cloud vertical structure for four distinct cloud regimes: daytime clear sky, daytime shallow convection, afternoon deep convection and nighttime convection (Figure 3 a-d). They are defined in as: (1) daytime clear sky day: precipitation rate = 0 mm day$^{-1}$ at all hours of the day and cloud fraction ≤ 5% at all levels between 0800 and 1600 local standard time (LST), (2) daytime shallow convection day: precipitation rate = 0 mm day$^{-1}$ at all hours of the day, and shallow cumulus clouds are identified by Berg and Kassianov (2008), who first selected cumulus clouds based on fine temporal resolution ARSCL data at ARM SGP and then manually scrutinized cloud images taken by the Total Sky Imager (available online at http://www.arm.gov/ instruments/tsi) to eliminate low cloud types other than shallow cumulus, (3) afternoon deep convection day: the diurnal maximum hourly precipitation rate ≥1 mm day$^{-1}$ and occurs between 1500 and 2000 LST and is at least twice more than the precipitation rate at any other hour of the day outside of 1500-2000 LST) and (4) nighttime deep convection day: the diurnal maximum hourly precipitation rate ≥1 mm day$^{-1}$ and occurs between 0000 and 0700 LST. Each composite consists of somewhere between 79 and 229 days spanning the warm seasons for a 10-year period. We also created the same model composites (Figure 3 e-h) from the model grid box nearest the SGP site (Figure 2), for the exact same days from the Day 2 hindcasts. We use the model cloud fraction for this comparison because the variables for using a radar simulator (Zhang et al. 2018; 2019) were not saved at the time the hindcasts were done. This analysis, which cannot be achieved from the usual AMIP simulations, is a more precise way of model parameterization evaluation because it minimizes the impact of erroneous large-scale states on the clouds. This is because the atmospheric large-scale state is closer to observations during each diurnal cycle of the hindcasts than it is in the AMIP simulation. Furthermore, multi-year hindcasts

provide a sufficient number of events to make a meaningful comparison with observations so that conclusions from such studies are more statistically robust.

In Figure 3, the model overestimates high clouds regardless of cloud regimes, even for clear sky condition. For clear sky condition, the model also shows middle- and low-level clouds. One possible explanation is that the deep convection scheme in the model is triggered whenever the convective available potential energy (CAPE) is larger than 70 J kg$^{-1}$. During the daytime in the warm seasons, CAPE is usually larger than the threshold and deep convection is easily triggered, resulting in the transport of water vapor and detrainment of cloud condensates. For the shallow convection regime, the model overestimates middle-level clouds by ~4-6% but underestimates shallow clouds by ~10%. For afternoon deep convective cloud regime, the model cannot simulate the transition from shallow to deep convective clouds. The deep convection clearly starts too early from around 11 local time rather than 15 in the afternoon. Also, the model underestimates both shallow and middle-level cloud fraction by ~10%. The model completely misses the nighttime convection regime, and only shows some deep convection starting around noon. The too early afternoon convection and the lack of nocturnal convection over land are common model problems as reported in previous studies (e.g., Dai 2006; Jiang et al. 2006; Covey et al. 2016). The missing nocturnal precipitation is likely related to the incapability of the model to capture elevated convection that often occurs at night at Southern Great Plains (Xie et al. 2019). We do realize that there are already small errors in the Day 2 large-scale state and they can also contribute to the errors in the simulated cloud fields. Nevertheless, their impact is still much smaller compared to the errors due to parameterization deficiencies in convection.

With multi-year hindcasts and long-term cloud observations to build up robust statistics, these comparisons help identify specific cloud regime deficiencies under very similar large-scale meteorological conditions, and model developers can further focus on improving specific processes represented in the cloud and convection parameterizations.

**3.2 Model biases associated with MJO**

The MJO (Madden and Julian 1971, 1972) is the dominant mode of intraseasonal variability in the tropics. MJO has significant impacts on the global water cycle as it can interact with many weather and climate phenomena (Zhang 2013). Nevertheless, contemporary GCMs still simulate poor MJO including its weak amplitude and lack of eastward propagation (Jiang et al.2015, Ahn et al. 2017). Recent studies suggest that the instability and propagation of the MJO are regulated by various feedback processes including cloud-radiation and wind-evaporation feedbacks (Sobel and Maloney 2012, 2013; Adams and Kim 2016; Ciesielski et al. 2017). These feedback processes may contribute to better MJO simulations if they are well represented in the GCMs. A particularly relevant process responsible for the eastward propagation of the MJO is the "pre-conditioning" process consisting of low-level moistening and shallow convective heating structure at the eastern edge of MJO deep convection (e.g., Jiang et al. 2011; Johnson and Ciesielski 2013; Powell and Houze 2013; Xu and Rutledge 2014). This process destabilizes the environment encouraging subsequent development of deep convection.

As each MJO event is unique from one another, one can take advantage of the multi-year hindcasts to composite precipitation, winds and diabatic heating profiles based on observed MJO phases with the focus on identifying robust model biases associated with the MJO. The diabatic heating rate or apparent heating of large-scale motion system (Q1) consists of the heating due to radiation, the release of latent heat by net condensation, and vertical convergence of the vertical eddy transport of sensible heat (Yanai et al. 1973). In CESM1/CAM5, Q1 can be calculated through summing up all the tendency terms of all the diabatic

processes. Figure 4 presents the observed composites of November to April 20-100 day band-pass filtered NOAA Interpolated Outgoing Longwave Radiation (OLR) anomalies and horizontal wind anomalies at 850mb from ERA-Interim, as a function of the eight phases of the MJO (Wheeler and Hendon 2004). The observed MJO shows a core of deep convection (center of negative OLR anomalies) over the Indian Ocean around 80°E associated with low-level convergence in winds during Phase 2. The core of deep convection slowly propagates eastward, and the intensity of convection decreases (with OLR anomalies

increasing) after the core of MJO crosses over the Maritime Continent and reaches the central Pacific (Phases 6-8). Figure 5 shows composites of November to April precipitation and horizontal wind biases from Day 3 hindcasts as a function of the eight phases of the MJO. We find that there is a dry bias in Day 3 hindcasts over the core of deep convection (center of negative OLR anomalies in Figure 4) associated with MJO, and a wet bias to the east over the region of suppressed convection (center of positive OLR anomalies in Figure 4) for all the phases as the MJO moves eastward. The dry bias is largest over Indian

Ocean during Phase 2 with magnitude ~ -6 mm day$^{-1}$, and the wet bias is largest over western Pacific during Phase 8 with magnitude ~5-6 mm day$^{-1}$. The dry bias is usually attributed to the lack of organized convection in the model (Moncrieff et al. 2017), and the wet bias is consistent with the too frequently triggered deep convection scheme even under suppressed large-scale condition. Further, there is a persistent dry bias over Borneo and part of Sumatra and wet bias around the Maritime Continent for all the phases indicating a possible local effect of diurnal cycle of convection. The dry bias is more significant

during Phase 4 and 5 as the MJO crosses over the Maritime Continent. The 850 mb winds show a biased low-level convergence near the Equator consistent with the excessive precipitation bias to the east over the region of suppressed convection.

During Phases 2 and 3 when the MJO is over the Indian Ocean, the anomalous Q1 profiles reveal that the magnitude of shallow heating is very weak (<0.4 K day$^{-1}$) to the east over the region of suppressed convection between 100°E and 120°E in Phase 2

and the heating is not restricted to low levels between 120°E and 150°E in Phase 3. Instead, there is an anomalous heating associated with deep convection in Phase 3, which is not evident in the observations as indicated from many previous studies (e.g., Figure 5a of Jiang et al. 2011). This suggests that the model fails to simulate the pre-conditioning moistening processes by shallow convection and the gradual transition from shallow to deep convection in Phase 3 of Day 3 hindcast composites. Figure 6 presents Hovmöller diagrams (longitude versus time in lag days) of rainfall anomalies along an equatorial band based

on the lag correlation over an Indian Ocean box for both GPCP and Day 3 hindcasts. The model shows an eastward propagation of precipitation anomalies associated with the MJO. However, the slightly lower correlation coefficients in the hindcasts (~0.3 vs ~0.4 in observation) east of 105°E is consistent with the weakening of MJO eastward propagation particularly over the

Maritime Continent. Previous studies have identified Maritime Continent as a "barrier" for MJO propagation and most MJO episodes in the models fail to propagate across it due to several possible reasons, such as interactions of convection, clouds, surface fluxes and local circulation within a diurnal cycle, land-sea contrast, terrain effect, or east-west low-level moisture gradient (Hagos et al. 2016; Jiang 2017; Zhang and Ling 2017). Although further diagnosis is required to fully understand the underlying problems for MJO propagation, our analysis with multi-year hindcasts indicates issues in the early development of biases in the shallow and deep convections associated with the MJO, as well as their interactions with the diurnal cycle of convection over the Maritime Continent.

While we examined the composites of the MJO phases using a pseudo time series from the multi-year hindcasts, caution should be exercised when interpreting the results with the discontinuous time series. Specifically, one should avoid examining those processes mentioned above within a single short hindcast. One could, however, perform longer hindcasts, like those in Klingaman et al., (2015) to investigate how model physics interact with the large-scale environment and influence the propagation and evolution of the MJO with longer lead times.

### 3.3 Interannual variations of moist processes

### 3.3.1 Variations of moist processes associated with ENSO

Being the leading mode of interannual variability in the tropics and extra-tropics, ENSO has significant impact on both regional and global temperature, circulation, and moist processes through teleconnections. To gain insights into whether or not errors in the response of these fields to SST anomalies can be attributed to parameterization errors or whether errors in the circulation response to SST anomalies also contribute, one can further contrast the multi-year hindcasts with the behavior of a companion AMIP simulation with the same boundary conditions (SST and sea ice). This question cannot be addressed before with one-year short-range hindcasts as we proposed in Ma et al. (2015). To this end, we first selected several fields to compute their monthly anomalies and then regressed these anomalous fields onto the Nino 3.4 index. Figure 7 shows the regression maps of precipitation, SWAbs, surface net flux (from atmosphere to the surface), and the surface zonal wind stress from observations, Day 2 hindcasts and the AMIP simulation (pattern statistics are shown in Table 3). The motivation for selecting these fields is because the tropical response of precipitation represents the atmospheric diabatic heating that forces circulation anomalies. On the other hand, surface radiation, turbulent heat fluxes, and wind stress provide critical heat and momentum forcings for SST anomalies and govern the ENSO behavior. The performance of these fields from an uncoupled atmospheric GCM is considered to be highly relevant for evaluation when it couples to an ocean model (Sun et al. 2006; Guilyardi et al. 2009).

The responses of these fields from Day 2 hindcasts show a better agreement both in the spatial patterns and magnitude with observations compared to the AMIP response (right column in Figure 7). This is especially evident for precipitation, absorbed shortwave flux and zonal wind stress over the Western North Pacific, South Pacific Convergence Zone and Indian Ocean. The

remote teleconnections may be chaotic or poorly done by the model, causing a poor simulation in the AMIP mode. The large-scale state is well constrained in the hindcasts and the response of those fields to SST anomalies in Figure 7 is much superior. This shows that remote errors are mostly the result of errors in circulation on long-term time scales although the errors in circulation may be caused by model physics in the first place and deteriorates through feedback processes with time. This is evident as there are still biases in the hindcasts indicating problems from parameterizations in representing those response to

SST changes even over local Nino 3.4 region.

Surface net flux and zonal wind stress also show a greater change between hindcast and AMIP response compared to precipitation and SWAbs. It is reasonable for the latter two moist processes to show less changes as they are fast processes and the biases associated with model parameterizations usually develop within a few days of model integration (Xie et al.

2012; Ma et al. 2014). It is also reasonable for zonal wind stress to show greater change as the low-level winds are well constrained for the hindcasts. For surface net heat flux, the errors are contributed from various flux terms including radiation and turbulent heat fluxes, which are affected by both model physics and dynamics. Therefore, the net heat flux shows the lowest spatial correlation and larger root mean square errors in both hindcasts and the AMIP simulation compared to other fields.

**3.3.2 Robustness of systematic errors**

One question raised from earlier studies (Xie et al. 2012 and Ma et al. 2014) of the correspondence between short- and long-time-scale errors is whether systematic errors of moist processes show significant interannual variation in the mean state biases. Figure 8 shows the pattern statistics between errors from the individual annual means in the hindcasts or AMIP simulation, and errors in the 16-year mean of the AMIP simulation (the reference fields) for precipitation, total cloud fraction (from the

ISCCP cloud simulator), SWAbs, and OLR. Compared to the long-term mean errors in the AMIP simulations, annual mean errors of the individual years for these fields show very similar magnitude in correlation and the normalized spatial standard deviation from the hindcasts at either time lag. This is also the case for individual AMIP years although the correlations and standard deviations show slightly larger spread. Compared to Day 2 hindcasts, the correlations and standard deviations from Day 3 hindcasts are closer to those from the AMIP simulations indicating the bias growth toward the AMIP bias with hindcast

lead time. We further find that the magnitude of correlations for annual mean errors between individual hindcast years and the long-term AMIP simulations are not sensitive to the ENSO phase in a given year for these fields. This is also the case if seasonal means are compared (Figures not shown here). These results suggest that mean errors in the moist processes are very robust and do not show significant interannual variations. Indeed, averaging the hindcast errors over many years (indicated by "2" or "3" in Figure 8) only slightly improves the agreement with the AMIP reference field. Thus, one may identify robust

model errors in the mean state from only one year of hindcasts with enough ensemble members (with ensemble members greater than 15, Ma et al. 2014). A similar conclusion with multiple years of short AMIP-type simulations was also suggested

by Wan et al. (2014). These results suggest that relatively short simulations will be effective at identifying the systematic moist process errors of a very high-resolution climate model which is too expensive to regularly perform multi-year simulations.

It is also of interest to compare the absolute magnitude of errors in individual years to that of the long-term systematic error in the AMIP simulation. To do so, we calculated the annually-averaged cloud error metrics proposed in Klein et al. (2013) in Figure 9. These metrics are scalar measures of performance in simulating the space-time distribution of several cloud measures, with better performance indicated by smaller E values. $E_{TCA}$ measures the error in total cloud amount, and $E_{CTP-\tau}$ measures the errors in the frequency of optically intermediate and thick clouds at high, middle, and low-levels of the atmosphere. $E_{SW}$ and

$E_{LW}$ measures the errors in the impacts on top-of-atmosphere shortwave and longwave radiation in the same cloud-top pressure and optical depth categories used for $E_{CTP-\tau}$, respectively. It is not surprising that the hindcasts show better performance in all the cloud metrics as the large-scale circulation and state are not too far from the reanalysis. This is also true for the interannual variations in global mean cloud radiative effect at the top of the atmosphere (Figure 10). We find that all the metrics and the cloud radiative effect show interannual variations indicating that the circulation and state anomalies make a significant

contribution to interannual variations although these metrics or errors in the cloud radiative effect are not sensitive to ENSO phase. We further find that there is a larger difference between hindcasts and AMIP in the total cloud amount error metric ($E_{TCA}$) implying that errors in the large-scale circulation and state make a larger contribution to errors in $E_{TCA}$ than cloud radiative properties (Figure 10).

## 4 Summary

In this study, we present a multi-year short-range hindcast experiment and its experiment design for better evaluating both the mean state and variability of atmospheric moist processes in climate models from diurnal to interannual time scales to facilitate model development. We also demonstrate that one can obtain unique understanding into robust GCM systematic moist processes errors by diagnosing these processes with corresponding observations for periods based on different phenomena. The present experiment also demonstrates that it is now feasible to systematically evaluate climate model moist processes in

deterministic weather-prediction mode just as the moist processes in weather prediction models are often evaluated in analyses or re-analyses (Jakob 1999, Yang et al. 2006). This experiment can also provide a very useful avenue to diagnose and understand critical processes regulating various climate and weather phenomena by taking advantage of detailed model output with a largely realistic representation of the large-scale state in hindcasts.

Three processes – the diurnal cycle of clouds during different cloud regimes at the ARM SGP Site, precipitation and diabatic heating associated with the MJO, and the response of moist processes to ENSO SST anomalies – are evaluated as examples of using this multi-year hindcast experiment to gain insights into robust model errors and their connection to physical parameterizations and large-scale state. These analyses can only be done through this multi-year hindcast experiment to establish robust statistics of the processes under well-controlled large-scale environment because these phenomena are either

interannual climate variability or only happen a few times in a given year (e.g. MJO, or cloud regime types). These comparisons identify specific model deficiencies that subsequent parameterization development should focus on. Results from the multi-year hindcasts also suggest that systematic errors in the mean state of moist processes are very robust and do not show significant interannual variation in error magnitude or patterns over large spatial domain. Although we only showed examples relevant to moist processes, other processes related to planetary boundary layer or radiation schemes can also be examined through this suite of experiments. The proposed experiment and evaluation method also complement the existing ways of climate model evaluation, such as performing GCM simulations in the AMIP, or nudging mode. Comparison among the multi-year hindcasts, AMIP and nudging simulations may provide more insights into these issues mentioned above.

In addition to processes indicated above, further studies on monsoon variability (e.g., South American and Asian monsoons, Chen et al. 2019), land-atmosphere interactions (Phillips et al. 2017), or detailed MJO studies with longer hindcast duration (Klingaman et al. 2015), are currently being explored with these hindcasts. As demonstrated in previous studies and here, model mean biases associated parameterized moist processes usually develop within a few days and manifest within weeks to affect the simulations of large-scale circulation and ultimately the climate mean state and variability. Therefore, model developers can achieve useful understanding of the underlying problems in model physics by conducting multiple years of hindcasts as demonstrated in the present work. Although newer version of the CAM and CLM is now available (CAM6/CLM5), similar systematic errors associated with moist processes remain present in the latest model version. Therefore, it is still worthwhile to study these hindcasts and compare the results to hindcasts with newer model version. In the meantime, we also plan to conduct another suite of multi-year hindcasts with the latest DOE Exascale Energy Earth System Model (E3SM, Golaz et al. 2019). We will also compare the results from E3SM to CESM1 to understand the impact of parameterization and model changes to the performance of moist processes since the atmospheric component of E3SM was originally branching form CAM5.3, which has very similar performance as CAM5 (Xie et al. 2018, Rasch et al. 2019). Note that E3SM version 1 has a new set of atmospheric physical parameterizations that are very similar to CAM6, the latest CAM. The hindcasts will also be available to the community once available.

Finally, the multi-year hindcast approach presented in this study is also intended as one of the experiment protocols which will be used in the Diurnal Cycle of Precipitation (DCP, https://portal.nersc.gov/cfs/capt/diurnal/) model intercomparison project under the Global Energy and Water cycle Exchanges (GEWEX) Global Atmospheric System Studies (GASS). This project is aimed to understand the processes that control the diurnal and sub-diurnal variations of precipitation over different climate regimes in observations and in models. The project will also identify the deficiencies and missing physics in current GCMs to gain insights for further improving the parameterization of convection.

**Code and data availability**

The model code is the CESM1 (cesm1_0_5, FC5 compset, F09_F09 resolution) and is available at http://www.cesm.ucar.edu/models/cesm1.0/. All model necessary input files are available at https://svn-ccsm-inputdata.cgd.ucar.edu/trunk/inputdata/. The boundary conditions of SST and sea ice data are available at
https://www.esrl.noaa.gov/psd/data/gridded/data.noaa.oisst.v2.html, The simulations are available online through the NERSC Science Gateways (https://portal.nersc.gov/archive/home/h/hyma/www/CAPT/CAPT_Long). The initial conditions are located at https://portal.nersc.gov/archive/home/h/hyma/www/CAPT/CAPT_Long/IC/. Detailed documentation for this experiment and variable list is at https://portal.nersc.gov/archive/home/h/hyma/www/CAPT/CAPT_Long/CAPT_Long_output_cesm1_0_5_v5.pdf.

**Author contribution**

HM designed, performed, analyzed the experiments, and wrote the first draft of the manuscript. SK and SX contributed to the design of the experiments and to the interpretation of the results. CZ, YZ, MZ, and XZ contributed to the analysis of experiments, and the interpretation of the results. WC, and CW contributed to the interpretation of the results. All coauthors contributed to the manuscript text.

**Acknowledgments**

This study was funded by the U.S. Department of Energy (DOE) Regional and Global Model Analysis program area, and the Atmospheric System Research and Atmospheric Radiation Measurement Programs. This work was performed under the auspices of the U.S. DOE by LLNL under contract DE-AC52-07NA27344.

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

**Table 1: Locations of model patch/site output associated with major field campaigns or DOE ARM sites.**

| Locations | Longitude | Latitude | grids |
|---|---|---|---|
| 1.Niamey-1 | 357E-359E | 14N-17N | 10 |
| 2.Niamey-2 | 0-9E | 5N-18N | 120 |
| 3.DYNAMO | 70E-83E | 10S-9N | 231 |
| 4.China-Shouxian | 114E-119E | 31N-34N | 25 |
| 5.Darwin | 129E-133E | 14S-10S | 20 |
| 6.Manus | 146E-149E | 4S-1S | 12 |
| 7.Nauru | 166E-169E | 2S-1N | 15 |
| 8.SHEBA | 190E-220E | 74N-78N | 125 |
| 9.MAGIC-1 | 201E-214E | 16N-25N | 121 |
| 10.MAGIC-2 | 207E-232E | 24N-30N | 147 |
| 11.MAGIC-3 | 221E-243E | 29N-36N | 162 |
| 12.NSA | 202E-206E | 70N-73N | 16 |
| 13.CARE | 238E-240E | 37N-39N | 9 |
| 14.SGP | 261E-264E | 35N-38N | 12 |
| 15.Vocals | 272E-291E | 23S-17S | 112 |
| 16.Amazonia | 296E-301E | 13S-9S | 25 |
| 17.Manaus | 298E-302E | 5S-1S | 25 |
| 18.Azores-Graciosa | 329E-334E | 38N-41N | 20 |
| 19.Barbados | 291E-301E | 10N-20N | 108 |




**Table 2: List of observation datasets.**

| Observations | Analyzed period | References |
|---|---|---|
| GPCP Precipitation V1.2 | 1997-2012 | Adler et al. 2003 |
| CERES EBAF Radiation Edition 2.8 | 2000-2012 | Loeb et al. 2009; Kato et al. 2013 |
| ISCCP D2 Cloud | 1997-2009 | Rossow et al. 1999 |
| ERA-Interim Reanalysis | 1997-2012 | Dee et al. 2011 |
| ARMBE ARSCL Cloud | 1997-2007 | Xie et al. 2010; Clothiaux et al. 2000, 2001 |






**Table 3. Pattern statistics of regression maps of selected fields onto the Niño 3.4 index between observations and model simulations (Day 2 hindcasts or AMIP).**

| | Spatial Correlation Coefficient | | RMSE | | Normalized Spatial Standard Deviation | |
|---|---|---|---|---|---|---|
| | Day 2 | AMIP | Day 2 | AMIP | Day 2 | AMIP |
| Precipitation | 0.94 | 0.81 | 0.25 | 0.44 | 1.02 | 0.97 |
| ISCCP Total Cloud Fraction | 0.83 | 0.68 | 2.11 | 2.84 | 0.91 | 0.82 |
| MODIS Total Cloud fraction | 0.85 | 0.72 | 2.12 | 2.74 | 1.03 | 0.94 |
| Absorbed Shortwave Radiation | 0.87 | 0.74 | 2.70 | 3.62 | 1.08 | 1.02 |
| Outgoing Longwave Radiation | 0.95 | 0.82 | 1.97 | 3.34 | 1.09 | 0.97 |
| Net Surface Fluxes | 0.79 | 0.61 | 4.42 | 6.19 | 0.90 | 0.98 |
| TAUX | 0.91 | 0.72 | 3.49 | 5.84 | 0.80 | 0.81 |
| OMEGA500 | 0.94 | 0.80 | 2.40 | 4.18 | 1.00 | 0.96 |
| U850 | 0.99 | 0.88 | 0.14 | 0.40 | 1.08 | 1.08 |
| U200 | 0.99 | 0.89 | 0.23 | 1.16 | 0.98 | 0.81 |


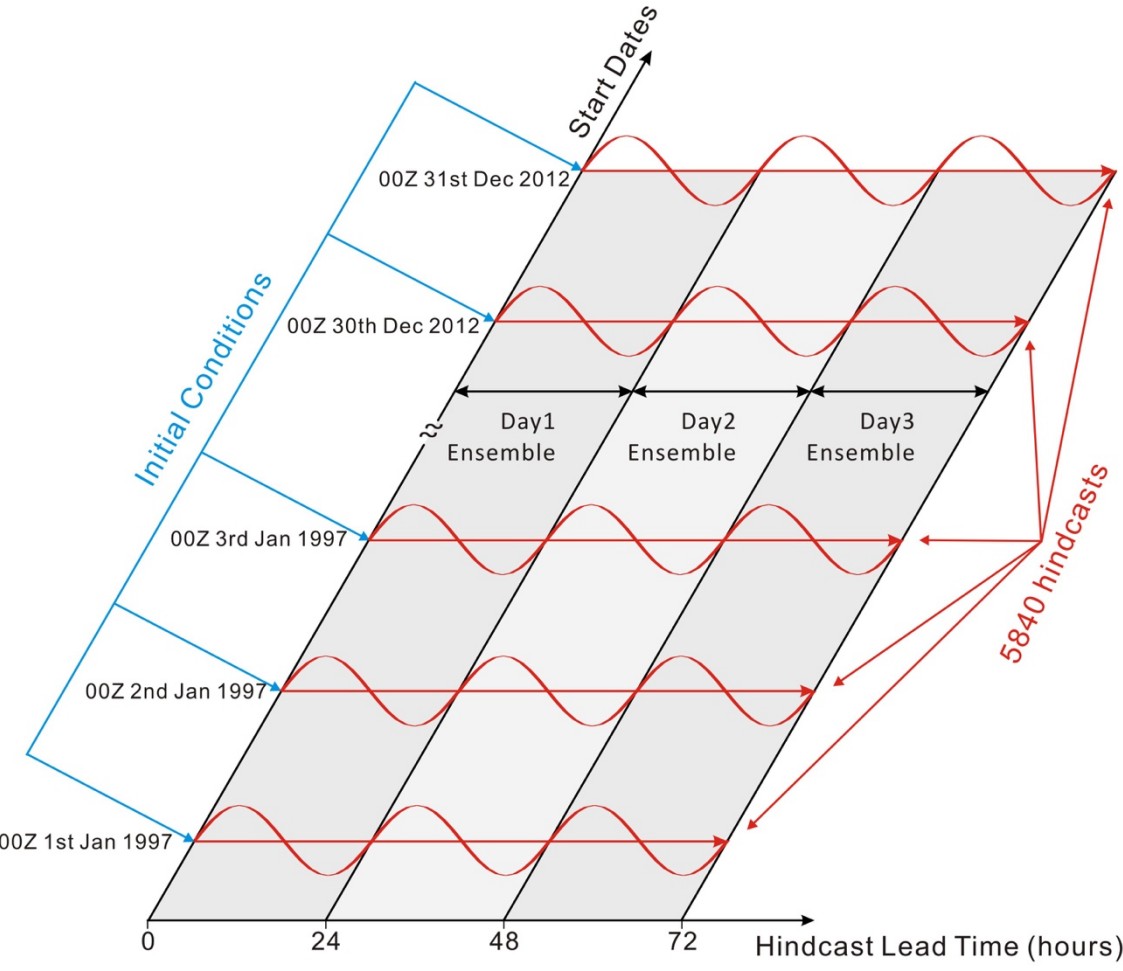

**Figure 1: Schematic diagram for the multi-year hindcast procedure (modified from Ma et al. 2015) applied to series of three-day hindcasts from 1997 to 2012. Each hindcast is initialized with ERA Interim Reanalysis and the starting time is 00Z every day.**

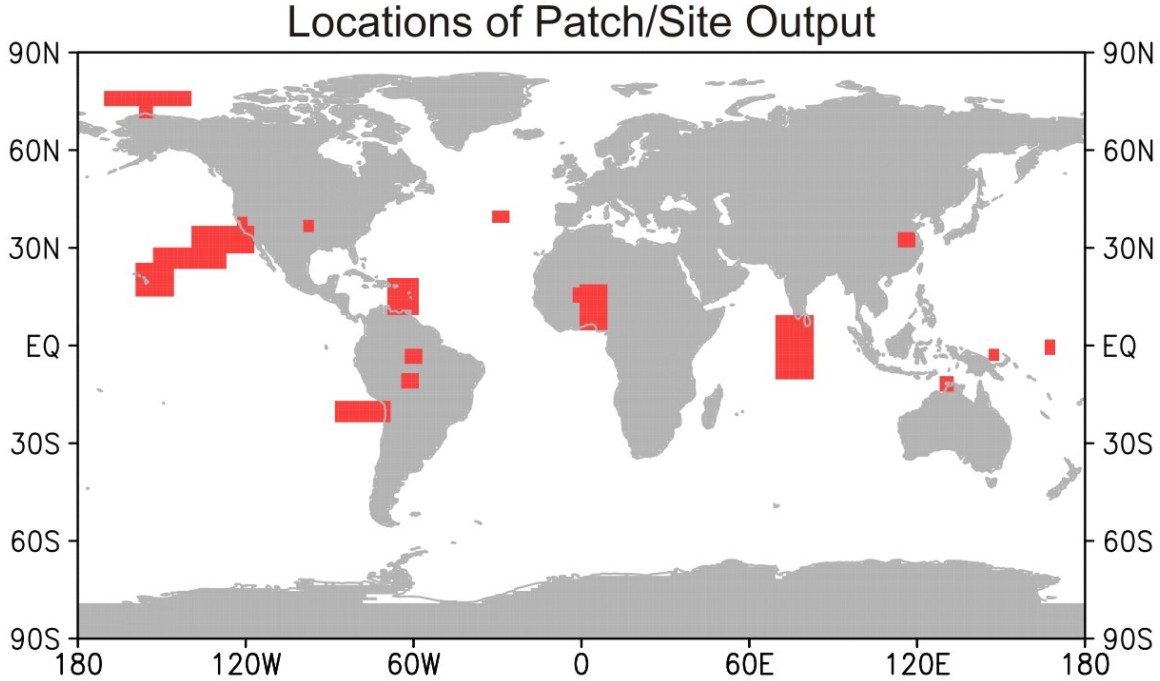

**Figure 2: Locations of model patch/site output associated with major field campaigns or U.S. Department of Energy Atmospheric Radiation Measurement (ARM) sites. See Table 1 for detailed longitude and latitude information.**

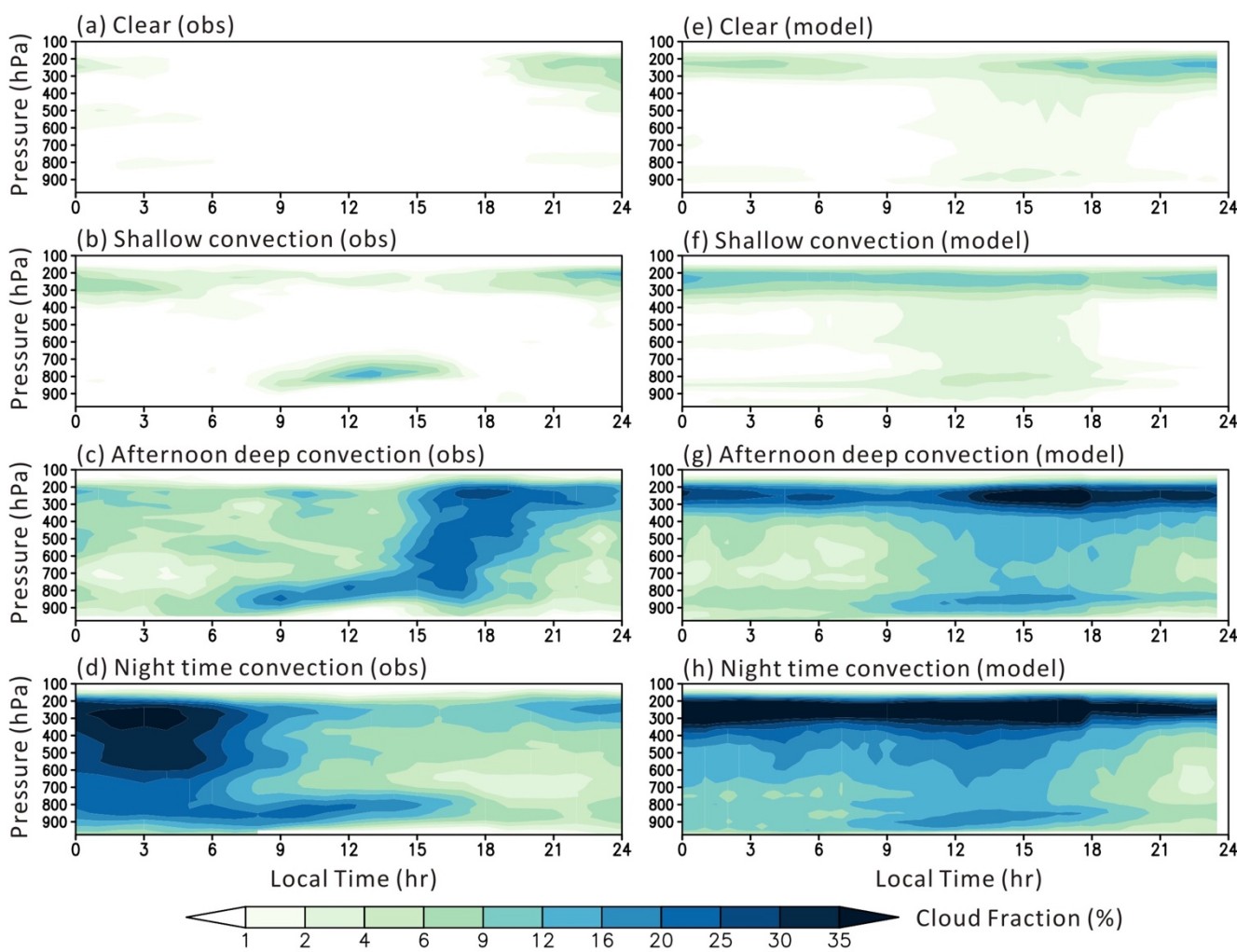

**Figure 3: Diurnal cycle of cloud fraction composites (%) from May to August in the years of 1997 to 2007 from (a)–(d) ARMBE ARSCL and (e)-(h) Day 2 hindcasts for different convection regimes: (a)(e) clear-sky regime, (b)(f) fair-weather shallow cumulus regime, (c)(g) late-afternoon deep convection regime, and (d)(h) nighttime deep convection regime. Figures (a)-(d) were modified from Figure 3 in Zhang and Klein (2010).**


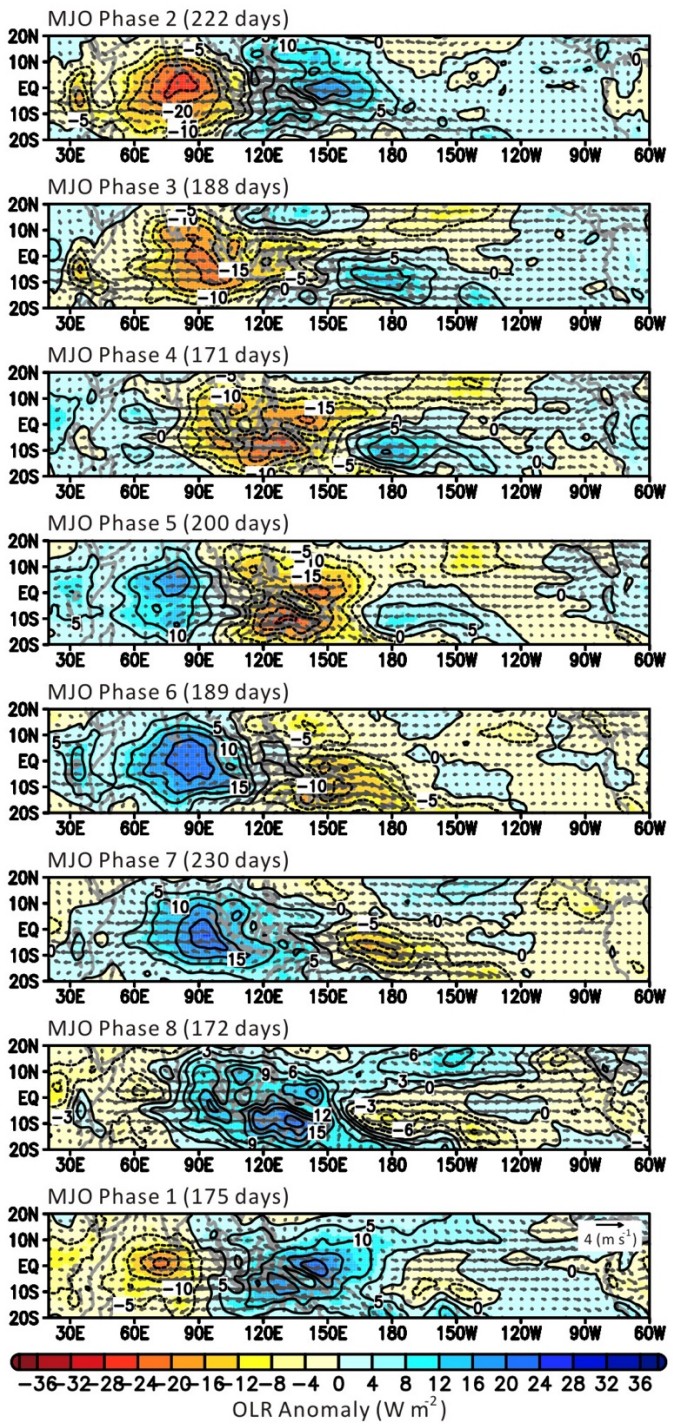

**Figure 4: Composites of November to April 20-100 day band-pass filtered NOAA Interpolated Outgoing Longwave Radiation anomalies (color shades and contours, W m$^{-2}$) and horizontal wind anomalies (vectors, m s$^{-1}$) at 850mb from ERA-Interim, as a function of the eight phases of the MJO (Wheeler and Hendon 2004). Number of days for composites are indicated on the top of each panel. Years of analysis are from 1997 to 2012.**

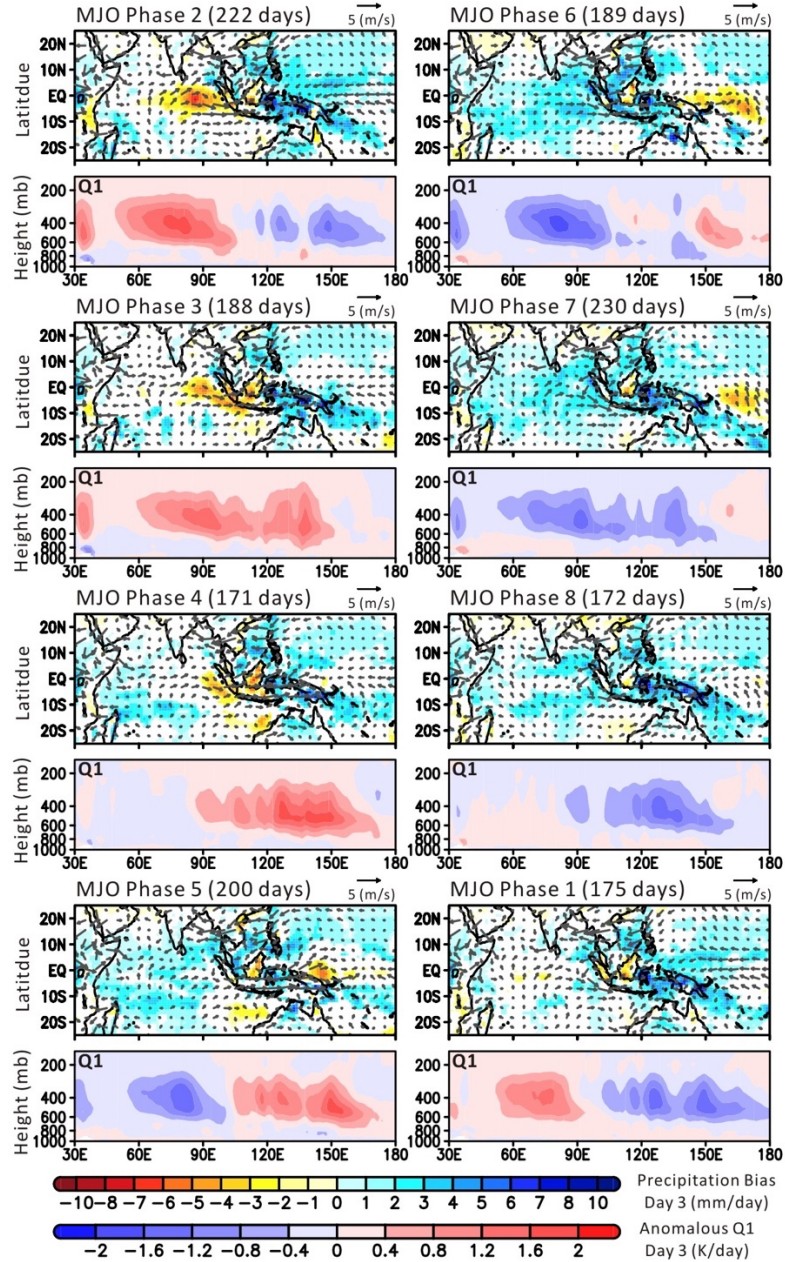


**Figure 5: Composites of November to April precipitation bias (color shades, mm day$^{-1}$), 850 mb horizontal wind biases (vectors, m s$^{-1}$), and anomalous 20-100 day band-pass filtered Q1 vertical profiles (averaged over 10ºS to 10ºN, K day$^{-1}$) from Day 3 hindcasts, as a function of the eight phases of the MJO (Wheeler and Hendon 2004). The number of days comprising each composite are indicated on the top of each panel. The observed precipitation and winds are from GPCP and ERA-Interim, respectively. Only**

**precipitation biases that are significant at the 95% confidence level are shaded. The Q1 profiles are computed directly from model's tendency terms with all the diabatic processes.**

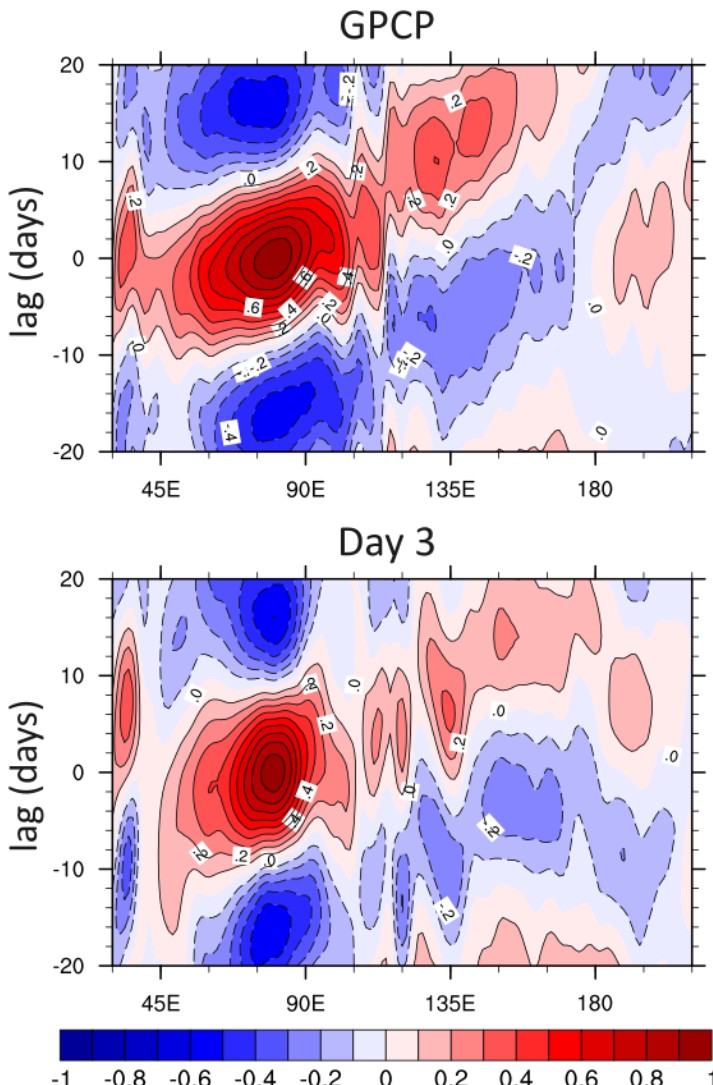

**Figure 6: Time-longitude anomalous rainfall correlation Homvöller diagram along the Equator (10ºS - 10ºN) based on GPCP and Day3 hindcasts. The rainfall anomalies associated with the MJO are derived based on lag-correlation over an Indian Ocean box (75º-85ºE; 5ºS-5ºN) for northern winter (November-April) of 1997 to 2012.**

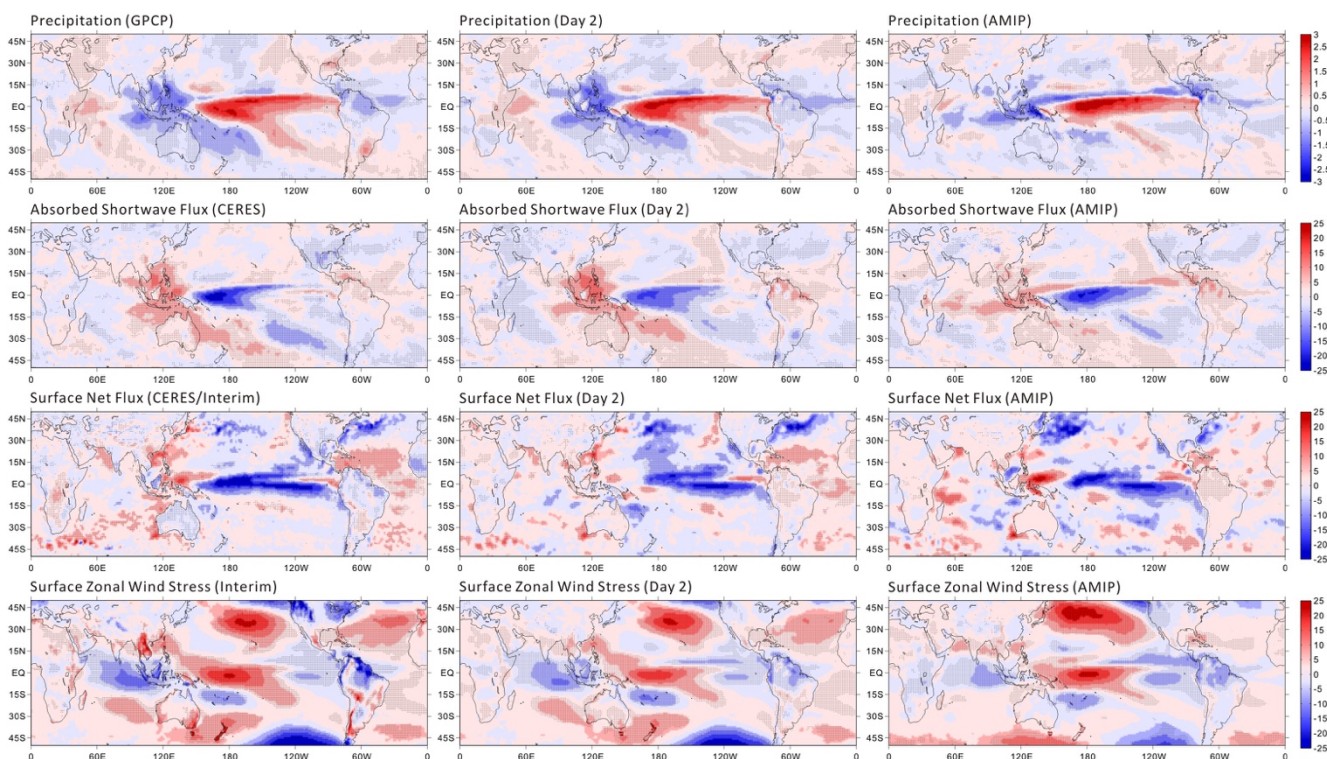

**Figure 7: Regression maps of precipitation (mm day$^{-1}$ K$^{-1}$), absorbed shortwave radiation (W m$^{-1}$ K$^{-1}$), net surface heat flux (W m$^{-1}$ K$^{-1}$), surface zonal wind stress (N m$^{-2}$ K$^{-1}$) onto the Niño 3.4 index from observations (left panels), Day 2 hindcasts (middle panels), and AMIP simulation (right panels).**


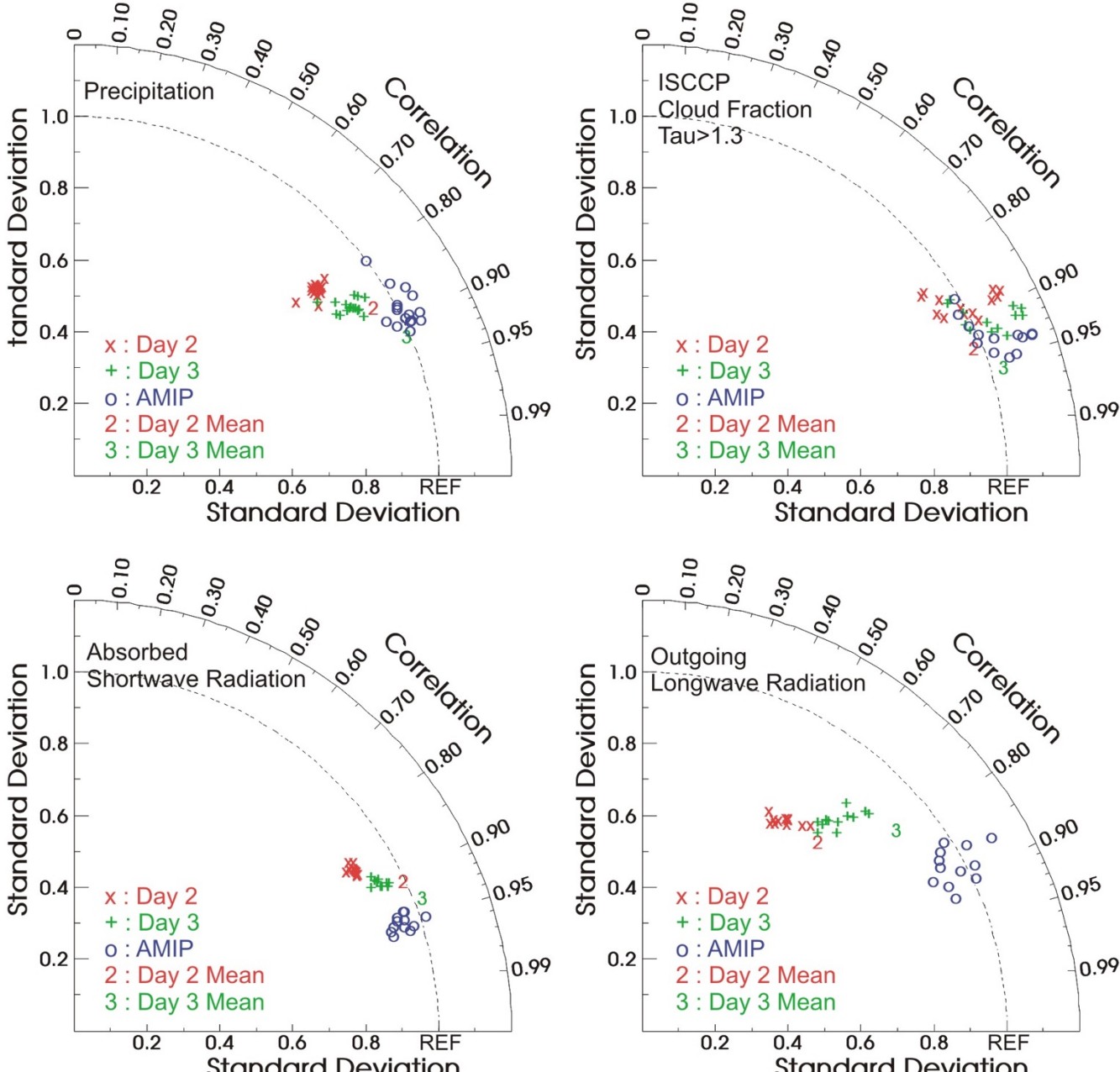

**Figure 8: Spatial pattern statistics of the bias errors (with respect to observations) of annual mean precipitation, ISCCP total cloud fraction (with tau > 1.3), absorbed shortwave radiation, and outgoing longwave radiation from the multi-year CAM5 hindcasts and an AMIP simulation. These statistics are illustrated with a Taylor diagram (Tayler et al. 2001) that shows the level of agreement of a given field to a common reference. The reference fields (REF) are the correspondent multi-year mean bias errors from the AMIP simulation. Each "x", "+", or "o" represents the Day 2, Day 3 or AMIP annual mean bias for individual years between 1997 and 2012 whenever the observations are available, and "2" and "3" represents the Day 2 and Day 3 hindcast mean biases averaged over all available years, respectively.**

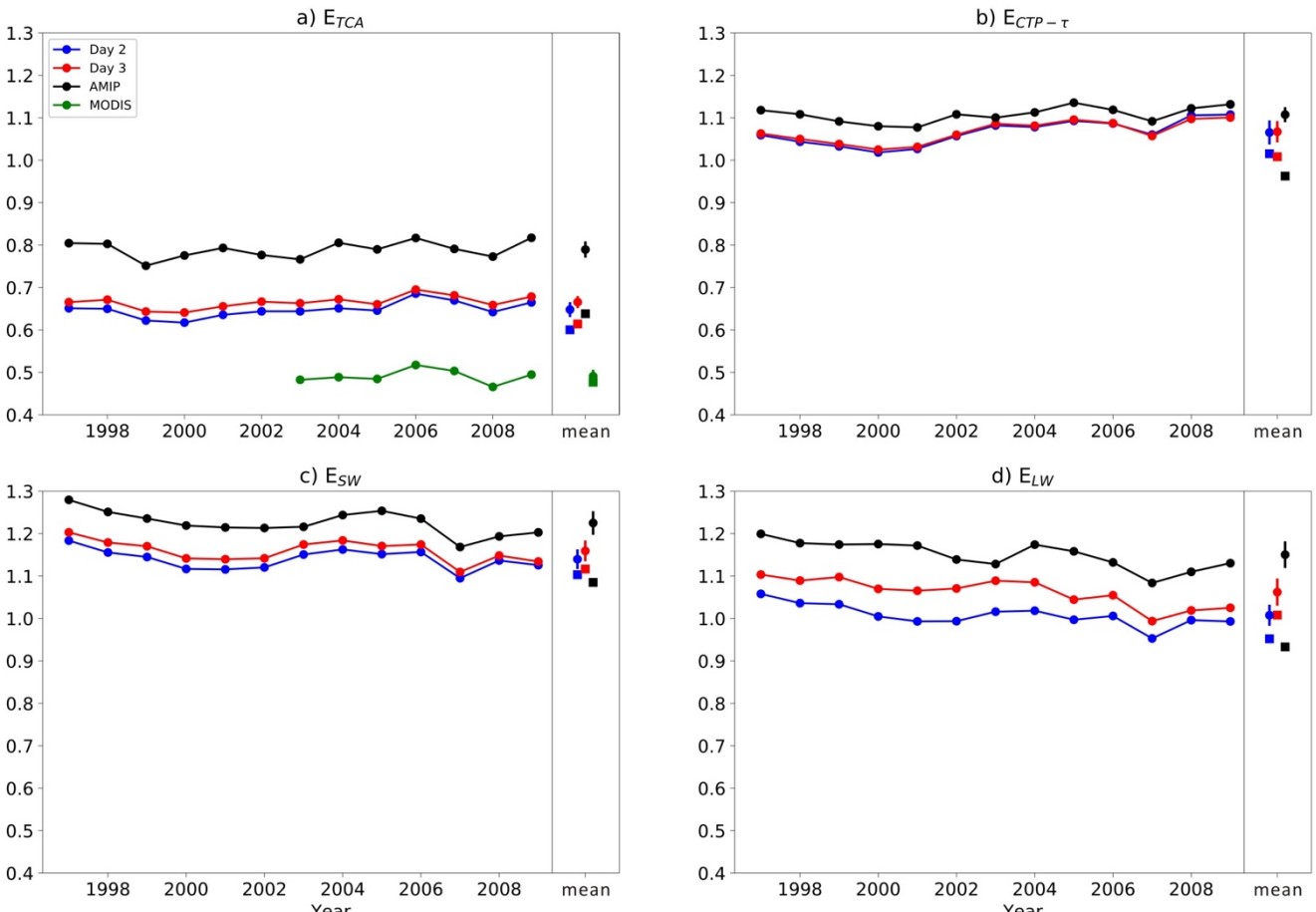

**Figure 9: Cloud metrics as defined using the ISCCP simulator in Klein et al. (2013) for Day 2 and 3 hindcasts, as well as the AMIP simulation. These metrics are scalar measures of performance in simulating the space-time distribution of several cloud measures, with better performance indicated by smaller E values. $E_{TCA}$ measures total cloud amount, and $E_{CTP\text{-}\tau}$ measures cloud-top pressure and optical depth in different categories of optically intermediate and thick clouds at high, middle, and low-levels of the atmosphere. $E_{SW}$ and $E_{LW}$ measure the impacts on top-of-atmosphere shortwave and longwave radiation in the same categories used for $E_{CTP\text{-}\tau}$, respectively. The markers with errorbars show the average and 1-σ interannual variation in these error metrics. The square symbols are the error metrics computed by comparing model and observed monthly-resolved climatological means. MODIS cloud amount for $E_{TCA}$ is plotted as a measure of observational uncertainty.**

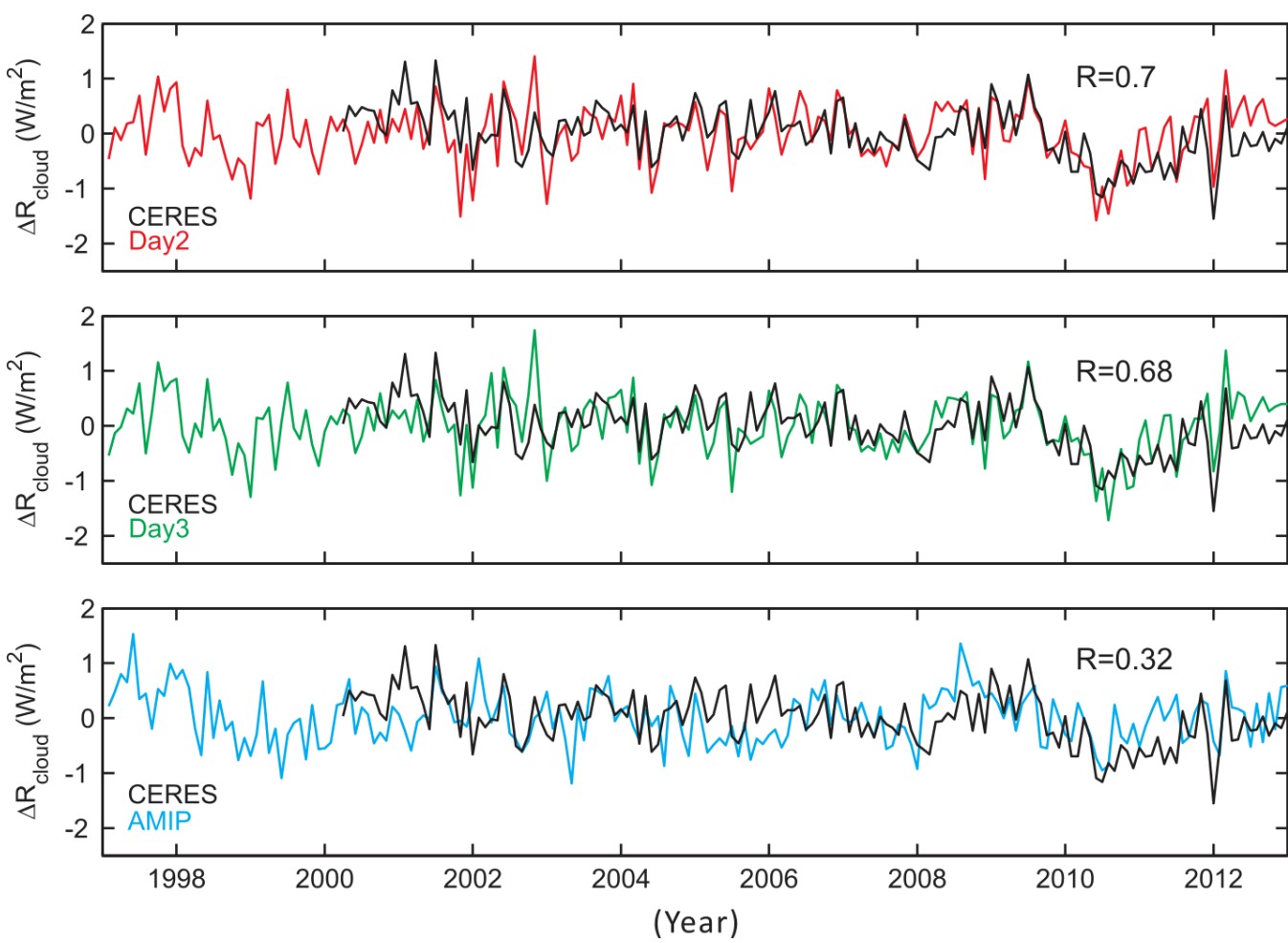

**Figure 10: Time series of monthly anomalies of global mean cloud radiative effect from CERES, Day 2 and 3 hindcasts, as well as from the AMIP simulation. The correlation coefficient between the model and observation is shown in the upper right portion of**
720 **each panel.**

725

