# Peer review of "A multi-year short-range hindcast experiment with CESM1 for evaluating climate model moist processes from diurnal to interannual time scales"

_Geoscientific Model Development, 2020_

## Referee Comment (RC1) · Anonymous Referee #1 · 4 May 2020

This study presents an innovative approach by conducting a series of short-range hindcasts based on a climate model that can be used to expose deficiencies in model parameterizations that are responsible for biases in climate simulations from the same model. Justification of this approach includes that model biases in climate mean and variability in long-term simulations start to emerge in the first several days of hindcasts by this model, which is thought to result from model parameterized processes since the large-scale state is still very close to the observations specified in the hindcast initial conditions. This study also proposed to conduct these short-range hindcasts for multiple years to achieve stable statistics when examining model deficiencies and associated processes, although it is shown that a systematic association between model

biases in short-range hindcasts and long-term simulations do not significantly change in different individual years. Three examples are further illustrated for applications of this approach to understand model biases in simulating the diurnal cycle of warm season precipitation over central U.S., the tropical MJO, and the local and remote influences by ENSO on the interannual time-scale. This approach along with the long-term hindcast datasets based on CAM5 produced from this project, can be used in many other studies for both tropical and extratropical climate phenomena, thus are valuable for the climate research community. In addition to identify model parameterization deficiencies as mentioned in this study, this approach can also provide a very useful avenue to diagnose and understand critical processes regulating various climate and weather phenomena by taking advantage of detailed model output with largely realistic representation of these phenomena in hindcasts at day-2. This can also be stressed in the manuscript.The paper is generally very clearly written although there is room for further improvement. I recommend this paper for publication after some minor revisions as listed below.

Minor comments: Line 12: suggest to change "3-day long hindcasts every day" to "3-day hindcasts initialized every day . . .". Also many grammatical errors need to be corrected throughout the manuscript, which I may not list all of them in the following. Line 14: suggest to delete "propagation", since MJO amplitude is also involved. Line 14: also not sure about "the responses of moist processes to sea surface temperature anomalies. . ." here. Why "moist processes" since net heat fluxes and wind stresses are also shown in Fig. 7? Line 21: again, in addition to "parameterized moist processes", could other parameterizations also related, for example, boundary layer and radiation parameterizations? Line 52: "AMIP" first appears here, but is defined in Line 99. Also, suggest to remove "of years". Line 57: CAM5 is mentioned here, but CESMv1 is mentioned in abstract and other places. Better be consistent. Line 66: suggest to delete "from diurnal to interannual time scales" in this line. Line 75: variation(s) Line 79: present(s) Line 96: of the 16-year duration Line 120-125: would be better if how these regimes are defined can be briefly explained. Line 135: cloud regime(s) Line

147: parameterization deficiencies Line 150: suggest change "scheme developer" to "model developer" Line 153: the intraseasonal variability Line 166: present(s) Line 170-171: "the intensity of OLR decreases" is a little confusing since active convection corresponds to smaller OLR. Not sure if this means convection is enhanced or weakened. Figure captions 3,6: CAPT are used but not defined. The presentation of Figures 4,5 can be improved, particularly Fig. 5. It may be not needed to use both precipitation and OLR since both of them represent MJO convection. At their present forms, these figures look very busy and a little difficult to see some detailed features particularly in wind vectors. Lines 179-180: Seems to me, the wet biases are largely evident over regions around the Maritime Continent, which may also relate to model biases in representing the regional diurnal cycle. Line 220: the circulation anomalies are discussed here, but temperature and moisture fields are also used to constrain model in the initial conditions. "the response SST anomalies is" needs to be fixed Line 231: "larger" root mean square errors? Line 254: Possibly to briefly mention these "cloud measures". Line 284-285: "Indeed, GCMs usually perform more poorly for …..". Not quite follow the purpose of this statement. Line 297: suggest change to "to understand the processes that control the diurnal and sub-diurnal variations of …." Line 301: CESM1 is mentioned here, and CAM5 in the model introduction part.

---

## Referee Comment (RC2) · Anonymous Referee #2 · 5 May 2020

**1   Reviewer's Summary of Manuscript**

In "A multi-year short-range hindcast experiment for evaluating climate model moist processes from diurnal to interannual time scales," Ma et al. describe an experimental design for (1) differentiating model errors that arise largely due to errors in parameterized processes (fast processes) versus errors that arise largely due to errors in the model's dynamical state (slow processes); and (2) for aiding in the identification of physical processes that might need focused model development to correct errors in fast processes. The experimental design consists of using a specific atmosphere-only GCM for two sets of simulations:

- 13 years of 3-day hindcast simulations, initialized using state information from ERA-Interim (interpolated to the model grid), and

- 13 years of an Atmospheric Model Intercomparison Project (AMIP) simulation

The authors implement this experimental protocol for the Community Earth System Model (CESM) version 1.0.5, using the CAM5 physics package. They compare simulations from the hindcast and AMIP simulations with a variety of observations, including ARM observations, reanalysis, satellite, and other. The analysis focuses on four aspects of the simulations: the diurnal cycle of convection over the southern Great Plains (SGP) of the United States; diabatic processes in the Madden-Julian Oscillation; dependence of diagnosed errors on El-Nino / Southern Oscillation (ENSO); and cloud radiative effects.

Their analysis of convection over the SGP shows that CESM1.0.5 has too much high cloud cover, which the authors speculate might be due to overly-frequent triggering of convection. They also show that the model has too little mid-level cloud coverage and too much shallow cloud coverage; possible reasons for this are not provided. Their analysis of the MJO shows that the model tends to have too little precipitation in the western half of the MJO core and too much precipitation on the eastern half, even in the hindcast simulations; and they suggest that diabatic processes associated with fast physical processes is likely the cause of these errors.

The authors show that the correlation between short- and long-term errors does not depend on the ENSO state, and they argue that their experimental design is therefore robust with respect to interannual variability. They also give evidence that errors in the large-scale circulation dominate errors in cloud radiative effects diagnosed from the model, relative to errors associated with fast physical processes (e.g., cloud parameterization) alone.

[Figure]

They conclude by arguing that this experimental design–and specifically the CESM 1.0.5 hindcast simulation output–are useful for model development.

**2  Summary of Review**

The authors do a nice job of summarizing several ways in which short-duration hindcast experiments can be used to diagnose the origins of errors in long-term climate simulations. Overall, the authors make a compelling argument that this type of experimental design is useful for decomposing errors into those associated with fast physical processes and slow (circulation-related) processes.

Despite this, I have several major reservations about the current form of the current manuscript, such that I cannot recommend this manuscript for publication at this time. Most critically, the current manuscript does not make a clear case for how this manuscript presents a new experimental design that is unique relative to what the authors have previously published on. Additionally, the authors do not provide enough information for a reader to replicate their experimental design. Finally, the authors' arguments are undermined by their use of a model version that has not been officially supported or developed for 7 years.

The following sections give a detailed description of these concerns, and the final section points out some specific issues that the authors should address if they submit a revised version of the manuscript.

**3 Major Concerns**

**3.1 Uniquness of manuscript and relevance for GMD**

The manuscript is presented as describing "a multi-year hindcast experiment and its experiment procedure" and giving "examples to demonstrate how one can better utilize simulations from this experiment design." Given that this is the authors' explicitly-stated premise for the paper, it is not immediately clear to me how this manuscript represents a unique contribution to the literature in general, and why it is a good fit for GMD in particular. The authors have written numerous papers about ensembles of short-duration hindcasts for over 15 years, and in that time they have already made a compelling argument that this experimental design is valuable. Why is yet-another-paper demonstrating the utility of this type of experimental design needed in the literature in general? After reading the manuscript several times, it is still not clear how this manuscript is unique relative to existing literature.

With respect to GMD in particular, there are three types of manuscript (wrt https://www.geoscientific-model-development.net/about/manuscript_types.html) that this could plausibly fit as:

1. Methods for assessment of models,

2. Model experiment description papers, or

3. Model evaluation papers.

If it is a manuscript of type #1, I believe the main contribution here would be "novel ways of comparing model results with observational data." The authors have written extensively about comparing hindcasts with ARM-SGP data (this was one of the earliest uses of CAPT, as far as I can tell), so the 1st example analyzed doesn't seem

sufficiently novel. The authors' composite analysis of MJO in hindcasts does seem unique in the literature, so this could considered a novel way of comparing models and observations. However, I would expect this analysis to be emphasized much more prominently in the paper if it was considered the main, novel contribution of the paper; instead, it is one of three examples that the authors run through, and the authors only devote three paragraphs to the topic. Finally, the analysis of diabatic processes associated with ENSO could be unique, but it wasn't clear to me what the authors were trying to show with this analysis, aside from showing that the correspondence between errors in short- and long-duration simulations is robust with respect to ENSO phase.

The way the manuscript is written, it seems that the authors are targeting manuscript type #2 "Model experiment description papers", in which case I would expect that the main contribution of the paper is "descriptions of standard experiments for a particular type of model," and "discussion of why particular choices were made in the experiment design and sample model output". The manuscript instead seems to describe a rather vague experimental protocol, consisting of short-duration hindcasts paired with AMIP simulations. The authors also describe their specific implementation of these hindcasts. It also isn't clear how the experimental design presented here differs substantially from other, seemingly similar experimental protocols described in the literature: e.g., Transpose-AMIP II (Williams et al., 2013), and the hindcast approach described by Ma et al. (2015). If it does differ, the authors should explicitly compare and contrast their proposed experimental protocol with those already described in the literature. In particular, Ma et al. (2015)–which this manuscript indicates the hindcast procedure is based on–states that "We also hope to provide guidance for those performing transpose-AMIP/CAPT simulations with their own climate models for model development or error diagnosis purposes." That sounds strikingly similar to a "model experiment description paper," which again raises the unanswered question of how this manuscript represents a unique contribution to the literature. The authors should also follow the GMD guidelines for this particular manuscript type, including giving the experimental protocol a name and version number in the manuscript title.

In its current form, I don't think this manuscript is appropriate to be considered a paper of type #3. The authors' analysis focuses on an outdated version of CESM that is no longer supported and is decreasingly used. Further, the current manuscript doesn't refer to any of the (extensive) existing literature evaluating CESM1, so there is currently no way to determine how/why this paper is unique relative to other model evaluation papers. (All of that said, I strongly suspect the authors weren't targeting this manuscript type, in which case such a literature review of CESM1 would not be needed.)

Williams, K.D., A. Bodas-Salcedo, M. Déqué, S. Fermepin, B. Medeiros, M. Watanabe, C. Jakob, S.A. Klein, C.A. Senior, and D.L. Williamson, 2013: The Transpose-AMIP II Experiment and Its Application to the Understanding of Southern Ocean Cloud Biases in Climate Models. J. Climate, 26, 3258–3274, https://doi.org/10.1175/JCLI-D-12-00429.1

Ma, H.-Y., C. C. Chuang, S. A. Klein, M.-H. Lo, Y. Zhang, S. Xie, X. Zheng, P.-L. Ma, Y. Zhang, and T. J. Phillips (2015), An improved hindcast approach for evaluation and diagnosis of physical processes in global climate models, J. Adv. Model. Earth Syst., 7, 1810–1827, doi:10.1002/ 2015MS000490.

**3.2 Repeatability of experiment**

Assuming that the authors are targeting this manuscript as a "Model experiment description paper", the manuscript does not appear to provide adequate detail for an external group to be able to replicate this experimental protocol a constrained way. More detail would need to be provided for the initial condition generation procedure, the nudging procedure, for the land-surface spinup procedure, and greenhouse gas and solar forcing (do these follow the AMIP protocol?). It also seems like it would be useful to guide other modeling groups on what variables should be saved as output: otherwise, how would experiments be intercompared?

Specifically, a number of questions come to mind that could impact the results from other modeling groups implementing this experiment:

- What method(s) should be used to horizontally and vertically remap the ERA-Interim data onto the model grid?

- Do any adjustments need to be made to the initial conditions to avoid spurious gravity waves associated with differences in topography between ERA-Interim and the given model?

- When running 'nudging' simulations to generate non-state-variable ICs:

  - What nudging method is used (this is provided in Ma et al. (2015), but it should be included here for completeness)?
  - How are the ERA-Interim data interpolated from the ERA-Interim times to the model's current time (nearest-neighbor, linear, spline, other?)
  - Is the nudging simulation run in an identical configuration to the AMIP run, with the exception of the nudging term?
  - Why aren't the land-surface conditions from the nudging simulation used for the land-surface initial condition in the hindcasts?

- When running the offline land-surface spinup simulation:

  - How are surface enthalpy and moisture fluxes calculated? (readers shouldn't have to read through old CESM documentation to figure out what the CESM-community-specific lingo for 'offline' land surface simulation refers to)
  - How should chemical fluxes be handled if needed by the land-surface model (e.g., C/N in the case of CLM5 in prognostic Carbon/Nitrogen mode)?

- – Why is the N. Viovy dataset used for forcing the offline land-surface spinup simulation?
- – How are data from these observational datasets interpolated to the model grid (nearest neighbor, linear, conservative remapping, other)?

The authors should provide the code that they use to generate the initial conditions, since there are undoubtedly numerous other questions about implementation of this experiment that would arise when external groups attempt to implement this protocol.

**3.3   Old model version**

The authors state that "model developers can achieve additional useful understanding of the underlying problems in model physics by conducting a multi-year hindcast experiment." However, this statement is undermined by the author's use of CESM 1.0.5, which stopped being supported and developed by NCAR years ago. The authors state "Although newer version [sic] of the CAM and CLM is now available (CAM6/CLM5), similar systematic errors associated with moist processes remain present in the latest model version," but they state this without any reference to manuscripts that support this statement. Further, CESM2 contains numerous upgrades to key parameterizations associated with moist processes: in particular the adoption of CLUBB, MG2 prognostic microphysics, and a retuning of the convection parameterization to "increase the sensitivity to convection inhibition". Because of all these changes, I don't see how the results from this dataset could be used to inform model development in CESM2, which is the only version of CESM under active development.

If this is indeed intended to be a "Model experiment description paper," then this point is somewhat less relevant. However, the authors should be more forthcoming about these caveats and the utility of the dataset produced as part of this paper. This paper would also be much more impactful if the authors made some specific comments about what

(if anything) would need to be done to implement this experimental design for CESM2.

**4 Specific, minor issues**

- pg 1, line 21: "associated parameterized" –> "associated with parameterized"

- pg 3, line 79: "Section 4 present" –> "Section 4 presents"

- pg 4, line 104: "output at model timestep" –> "output for every model timestep" (?)

- pg 5, lines 156-158: this is one of several theories for the MJO (e.g., see Yang and Ingersoll, 2011), so these feedback processes may not all be necessary

- pg 6, line 165: what is Q1?

- pg 7, line 220: "the response SST anomalies is much superior" –> "the response to SST anomalies is much superior"

- pg 7, line 221: "the result of poor circulation" –> "the result of errors in circulation" ('poor circulation' is usually reserved for describing anatomical difficulties with blood flow)

- pg 8, line 253: "the annualy cloud error metrics" –> "the annually-averaged cloud error metrics" (?)

- pg 9, lines 276-277: "These comparisons identify. . . " this statement only makes sense to include if the authors repeat the experiments with an actively-developed model version.

Yang, D. and A.P. Ingersoll, 2011: Testing the Hypothesis that the MJO is a Mixed Rossby–Gravity Wave Packet. J. Atmos. Sci., 68, 226–239, https://doi.org/10.1175/2010JAS3563.1

————————————————————————

---

## Referee Comment (RC3) · Anonymous Referee #3 · 9 Jun 2020

Review of GMD-2020-39

A multi-year short-range hindcast experiment for evaluating climate model moist processes from diurnal to interannual time scales

General comments:

The authors describe a multi-year hindcast experiment, in which short (3-day) initialised forecasts are performed each day with a climate model. The aim of the experiment is to understand the systematic errors associated with moist processes, which typically appear within the first few days of a model simulation. By performing short forecasts daily for 16 years, the authors are able to generate robust statistics of these systematic

errors. This allows them to understand whether the errors depend on the phase of natural variability, or to aggregate statistics across all forecasts to minimise the effects of that natural variability. The authors provide three examples of the types of analysis that could be performed with this hindcast set: the diurnal cycle of clouds over the central United States, the propagation of the Madden-Julian Oscillation through the Maritime Continent and the response of tropical precipitation and circulation to the El Nino-Southern Oscillation.

While I believe that the multi-year hindcast experiment is a useful framework to understand model systematic errors, it is not clear that this manuscript advances either the experiment protocol itself or the methods used to analyse those experiments. The authors have performed and analysed similar experiments in the past, as have other groups. The analysis and conclusions presented here are brief, mainly descriptive, and occasionally erroneous. The authors do not consistently show how the errors diagnosed in the short-range hindcasts differ from those diagnosed from AMIP experiments, which is important to demonstrate the value of their framework and to understand the AMIP errors. The interpretation of MJO propagation in such short simulations is also problematic. Finally, the authors motivate their experiment framework by invoking process-level improvements in models, but it is not clear how the analysis shown here would directly inform specific, targeted model development efforts to reduce these biases. I expand on these points below.

Major comments:

1. This manuscript is a GMD "model experiment description paper", which are "descriptions of standard experiments for a particular type of model". This manuscript is missing many of the GMD criteria for such a paper, including a name for the experiment, a version number for the protocol, and a version number for the boundary and initial conditions. The description of the experiment design in the paper also falls short of expectations; it lacks much of the detail that one would find in the description for a MIP, for instance, which seems to be the GMD gold standard. In particular, the following

details are not clear:

a. Do all initial conditions come from the nudged simulations, or do some fields come directly from ERA-Interim? The current description at L86 is not clear on this. b. What fields are nudged in the nudging simulations? c. What reanalysis and observation datasets were used force the offline land simulation? d. Is land initialised to a climatology from the offline simulations, or to the value on a particular day? If the latter, is the value taken from the last spinup cycle of the land model? e. Do greenhouse gases and aerosols follow the AMIP specification?

Also, I note that the offline land simulation data are not published, according to the citation in the manuscript, but GMD specifies that all boundary conditions must be published under version control, as part of the paper.

2. The novel contribution of the manuscript to the design and analysis of short-range hindcast experiments is not clear. The authors have conducted similar experiments in the past through the CAPT project (e.g., Ma et al., 2015); other similar experiment designs exist, such as Transpose-AMIP. This is far from the first paper to propose such an approach, which I think the authors would acknowledge. What is new and innovative about this experiment design? How does this experiment design enable new understanding of the development of systematic errors related to moist processes, beyond that which could be achieved through existing protocols?

3. At various points in the paper (e.g., L129, L155, L277), the authors motivate their analysis of short-range hindcast simulations by invoking parameterisation development or parameterisation evaluation. Yet, it is difficult to see how a parameterisation developer could gain useful insight from the simulations the authors present. If I were developing a convection parameterisation, I do not know how I would be able to apply the authors' analysis to improve that parameterisation. The authors do not test the sensitivity to the choice of model parameterisation, let alone to choices of particular model parameters. To make such a test would require repeating the full hindcast set,

potentially several times to test the sensitivity to various choices. I would expect that this would be costly, both in computational and human resources. Is there actionable information for parameterisation improvement here? Can the authors point to specific parameterisation developments that were made in response to their previous work with short-range hindcasts?

4. The authors state that their aim is to understand which of the errors seen in AMIP simulations are due to local-scale errors in moist processes, and which are due to errors in larger-scale or remote processes (e.g., L274-275). However, for the cloud regimes analysis (section 3.1) and the MJO propagation analysis (section 3.2), the authors analyse only the short-range hindcasts, without reference to the AMIP simulation. Thus, it is not possible to understand the relative contributions of moist-process errors. I suggest adding similar analysis for the AMIP simulation, as the authors have done for the ENSO analysis (section 3.3). Otherwise, the value of the short-range hindcasts over the AMIP simulations is not particularly clear.

5. The MJO analysis seems incomplete and problematic. In particular:

a. The authors show diabatic heating (Q1) profiles from the model, but do not compare them against observations (e.g., satellite-derived heating profiles). Yet, they make statements that the model heating is "very weak" (L186) and "not restricted to low levels". This suggests a bias, but the reader cannot judge the bias as there is no truth against which to compare!

b. The authors discuss MJO "propagation" through the Maritime Continent (L195), but in such short (3-day) hindcasts there is no "propagation" as such. The model is constantly reinitialised from the nudged simulations, so no MJO event could ever propagation across the Maritime Continent in a single hindcast. Likewise, the authors discuss a lack of "pre-conditioning" and "gradual transition from shallow to deep convection" in the model (L189). These are concepts often used in free-running climate simulations; the degree to which they apply to such short hindcasts is not clear. There can be no

"pre-conditioning" or "gradual" transition in the span of a single hindcast.

c. Related to the above, it may be useful to diagnose how well the MJO propagates in the nudged simulation used as initial conditions. I believe comparisons of the nudged simulation against the hindcasts may provide more insight into how much of the lack of MJO propagation is due to errors in moist processes in the hindcasts.

d. The authors mention errors in "interactions with the diurnal cycle of convection over the Maritime Continent" (L200), related to the MJO propagation errors, but provide no evidence or analysis of these.

6. For the ENSO analysis (section 3.3), it is not clear why the authors analyse the ENSO regressions globally, given that the authors are interested in the fast (1-3 day) response to ENSO SST anomalies. The far-field response (e.g., over the Indian Ocean or the Atlantic) takes a few weeks to develop, at least. Thus, I suspect that the errors in these regions are not errors in the response to Pacific SST anomalies in the ENSO region, but rather are errors in response to the local circulation and SST, which may or be not be directly related to ENSO. Perhaps I am missing something, but the interpretation of the analysis here seems to be less straightforward than the authors suggest. Also, related to the ENSO section:

a. L210: The authors say that there are statistics in Table 2, but I cannot see any! Where are the pattern statistics that the authors mention?

b. At several places, (e.g., L216, L220), the authors state the the hindcasts are "superior", or have "better agreement" with observations, when compared to the AMIP simulations. This agreement is not obvious, particularly in such small figures. It needs to be quantified statistically.

7. In Figure 8, why are the OLR pattern correlations much lower than for the other variables?

8. L247: The authors suggest that robust model errors can be identified "from only one

year of hindcasts with enough ensemble members." How many is enough? Can the authors estimate the number of ensemble members required from their simulations?

9. L249: "short simulations will be effective at reducing moist process errors" - Simulations alone do not reduce errors! Only dedicated model development efforts can reduce these errors. How does this framework, and the results the authors show, contribute to this effort? See also comment 1 above.

---

## Author Comment (AC1) · 9 Jul 2020

Anonymous Referee #1 This study presents an innovative approach by conducting a series of short-range hindcasts based on a climate model that can be used to expose deficiencies in model parameterizations that are responsible for biases in climate simulations from the same model. Justification of this approach includes that model biases in climate mean and variability in long-term simulations start to emerge in the first several days of hindcasts by this model, which is thought to result from model parameterized processes since the large-scale state is still very close to the observations specified in the hindcast initial conditions. This study also proposed to conduct these

short-range hindcasts for multiple years to achieve stable statistics when examining model deficiencies and associated processes, although it is shown that a systematic association between model biases in short-range hindcasts and long-term simulations do not significantly change in different individual years. Three examples are further illustrated for applications of this approach to understand model biases in simulating the diurnal cycle of warm season precipitation over central U.S., the tropical MJO, and the local and remote influences by ENSO on the interannual time-scale. This approach along with the long-term hindcast datasets based on CAM5 produced from this project, can be used in many other studies for both tropical and extratropical climate phenomena, thus are valuable for the climate research community. In addition to identify model parameterization deficiencies as mentioned in this study, this approach can also provide a very useful avenue to diagnose and understand critical processes regulating various climate and weather phenomena by taking advantage of detailed model output with largely realistic representation of these phenomena in hindcasts at day-2. This can also be stressed in the manuscript. The paper is generally very clearly written although there is room for further improvement. I recommend this paper for publication after some minor revisions as listed below.

Response to reviewer:

We thank the reviewer for all the comments. Those certainly helped to improve our manuscript. A pdf supplemental file of this response is also available in the link at the end.

We have stressed the point raised by the review regarding "This experiment can also provide a very useful avenue to diagnose and understand critical processes regulating various climate and weather phenomena by taking advantage of detailed model output with a largely realistic representation of the large-scale state in hindcasts" in the revised manuscript over Lines 339-341 in the Summary Section.

Please see our point-by-point responses to the minor moments below.

Minor comments: Line 12: suggest to change "3-day long hindcasts every day" to "3-day hindcasts initialized every day . . .". Also many grammatical errors need to be corrected throughout the manuscript, which I may not list all of them in the following.

Response to reviewer: The sentence is revised as suggested and we have carefully checked the grammatical errors throughout the entire manuscript.

Line 14: suggest to delete "propagation", since MJO amplitude is also involved.

Response to reviewer: Propagation deleted as suggested.

Line 14: also not sure about "the responses of moist processes to sea surface temperature anomalies. . ." here. Why "moist processes" since net heat fluxes and wind stresses are also shown in Fig. 7?

Response to reviewer: We changed the sentence to "the response of precipitation, surface radiative and heat fluxes, as well as zonal wind stress to sea surface temperature anomalies associated with the El Niño-Southern Oscillation - . . ." to better matched what is shown in Figure 7.

Line 21: again, in addition to "parameterized moist processes", could other parameterizations also related, for example, boundary layer and radiation parameterizations?

Response to reviewer: Boundary layer and radiation parameterizations are certainly related based on our previous studies and experience. We did not mention these two in the abstract because we didn't show any results directly linked to these two schemes. In the revised manuscript, we added the information "Although we only showed examples relevant to moist processes, other processes related to planetary boundary layer or radiation schemes can also be examined through this suite of experiments." in the summary section over Lines 337-339.

Line 52: "AMIP" first appears here, but is defined in Line 99. Also, suggest to remove "of years".

Response to reviewer: AMIP is now defined when it first appears in the introduction, and "of years" is removed as suggested.

Line 57: CAM5 is mentioned here, but CESMv1 is mentioned in abstract and other places. Better be consistent.

Response to reviewer: We now changed it to CESM1 to be consistent across the manuscript.

Line 66: suggest to delete "from diurnal to interannual time scales" in this line.

Response to reviewer: Deleted as suggested.

Line 75: variation(s)

Response to reviewer: Revised.

Line 79: present(s)

Response to reviewer: Revised.

Line 96: of the 16-year duration

Response to reviewer: Revised.

Line 120-125: would be better if how these regimes are defined can be briefly explained.

Response to reviewer: The cloud regimes are now defined and briefly explained following Zhang and Klein (2010) in the revised manuscript over Lines 160-169.

Line 135: cloud regime(s)

Response to reviewer: Revised.

Line 147: parameterization deficiencies

Response to reviewer: Revised.

Line 150: suggest change "scheme developer" to "model developer"

Response to reviewer: Revised as suggested.

Line 153: the intraseasonal variability

Response to reviewer: Revised.

Line 166: present(s)

Response to reviewer: Revised.

Line 170-171: "the intensity of OLR decreases" is a little confusing since active convection corresponds to smaller OLR. Not sure if this means convection is enhanced or weakened.

Response to reviewer: The sentence is revised to "the intensity of convection decreases (with OLR anomalies increasing) after the core of MJO crosses over the Maritime Continent" to avoid the confusion.

Figure captions 3,6: CAPT are used but not defined.

Response to reviewer: We now removed CAPT from the figure captions.

The presentation of Figures 4,5 can be improved, particularly Fig. 5. It may be not needed to use both precipitation and OLR since both of them represent MJO convection. At their present forms, these figures look very busy and a little difficult to see some detailed features particularly in wind vectors.

Response to reviewer: We have removed OLR contours in Figure 5 to make the figure less busy in the revised manuscript.

Lines 179-180: Seems to me, the wet biases are largely evident over regions around the Maritime Continent, which may also relate to model biases in representing the regional diurnal cycle.

Response to reviewer: We agree that the wet biases may also relate to the biases in

the diurnal cycle of the Maritime Continent. We now revised the sentence to "Further, there is a persistent dry bias over Borneo and part of Sumatra and wet bias around the Maritime Continent for all the phases indicating a possible local effect of diurnal cycle of convection." in the revised manuscript over Lines 228-229.

Line 220: the circulation anomalies are discussed here, but temperature and moisture fields are also used to constrain model in the initial conditions. "the response SST anomalies is" needs to be fixed

Response to reviewer: We revised the sentence to "The large-scale state is well constrained in the hindcasts and the response of those fields to SST anomalies in Figure 7 is much superior".

Line 231: "larger" root mean square errors?

Response to reviewer: "larger" is added in the revised manuscript.

Line 254: Possibly to briefly mention these "cloud measures".

Response to reviewer: We now added the following sentences "ETCA measures the error in total cloud amount, and ECTP-t measures the errors in the frequency of optically intermediate and thick clouds at high, middle, and low-levels of the atmosphere. ESW and ELW measures the errors in the impacts on top-of-atmosphere shortwave and longwave radiation in the same cloud-top pressure and optical depth categories used for ECTP- t, respectively" in the revised manuscript.

Line 284-285: "Indeed, GCMs usually perform more poorly for . . ...". Not quite follow the purpose of this statement.

Response to reviewer: We have removed this sentence in the revised manuscript.

Line 297: suggest change to "to understand the processes that control the diurnal and sub-diurnal variations of . . .."

Response to reviewer: This sentence is revised as suggested.

Line 301: CESM1 is mentioned here, and CAM5 in the model introduction part.

Response to reviewer: We now use CESM1 across the manuscript.

Please also note the supplement to this comment:
https://gmd.copernicus.org/preprints/gmd-2020-39/gmd-2020-39-AC1-supplement.pdf

---

## Author Comment (AC2) · 9 Jul 2020

**1 Reviewer's Summary of Manuscript**

In "A multi-year short-range hindcast experiment for evaluating climate model moist processes from diurnal to interannual time scales," Ma et al. describe an experimental design for (1) differentiating model errors that arise largely due to errors in parameterized processes (fast processes) versus errors that arise largely due to errors in the model's dynamical state (slow processes); and (2) for aiding in the identification of physical processes that might need focused model development to correct errors in fast processes. The experimental design consists of using a specific atmosphere-only GCM for two sets of simulations:

• 13 years of 3-day hindcast simulations, initialized using state information from ERA-Interim (interpolated to the model grid), and

• 13 years of an Atmospheric Model Intercomparison Project (AMIP) simulation

The authors implement this experimental protocol for the Community Earth System Model (CESM) version 1.0.5, using the CAM5 physics package. They compare simulations from the hindcast and AMIP simulations with a variety of observations, including ARM observations, reanalysis, satellite, and other. The analysis focuses on four aspects of the simulations: the diurnal cycle of convection over the southern Great Plains (SGP) of the United States; diabatic processes in the Madden-Julian Oscillation; dependence of diagnosed errors on El-Nino / Southern Oscillation (ENSO); and cloud radiative effects.

Their analysis of convection over the SGP shows that CESM1.0.5 has too much high cloud cover, which the authors speculate might be due to overly-frequent triggering of convection. They also show that the model has too little mid-level cloud coverage and too much shallow cloud coverage; possible reasons for this are not provided. Their analysis of the MJO shows that the model tends to have too little precipitation in the western half of the MJO core and too much precipitation on the eastern half, even in the hindcast simulations; and they suggest that diabatic processes associated with fast physical processes is likely the cause of these errors.

The authors show that the correlation between short- and long-term errors does not depend on the ENSO state, and they argue that their experimental design is therefore robust with respect to interannual variability. They also give evidence that errors in the large-scale circulation dominate errors in cloud radiative effects diagnosed from the model, relative to errors associated with fast physical processes (e.g., cloud parameterization) alone.

They conclude by arguing that this experimental design–and specifically the CESM 1.0.5 hindcast simulation output are useful for model development.

**2 Summary of Review**

The authors do a nice job of summarizing several ways in which short-duration hindcast experiments can be used to diagnose the origins of errors in long-term climate simulations.

Overall, the authors make a compelling argument that this type of experimental design is useful for decomposing errors into those associated with fast physical processes and slow (circulation-related) processes.

Despite this, I have several major reservations about the current form of the current manuscript, such that I cannot recommend this manuscript for publication at this time. Most critically, the current manuscript does not make a clear case for how this manuscript presents a new experimental design that is unique relative to what the authors have previously published on. Additionally, the authors do not provide enough information for a reader to replicate their experimental design. Finally, the authors' arguments are undermined by their use of a model version that has not been officially supported or developed for 7 years.

The following sections give a detailed description of these concerns, and the final section points out some specific issues that the authors should address if they submit a revised version of the manuscript.

*Response to reviewer:*
*We thank the reviewer for all the comments. Those certainly helped to improve our manuscript.*

*Regarding the new uniqueness of the current experiment, this is our first time documenting this suite of multi-years (1997-2012, 16 years) hindcasts and proposing on how one can better utilizing these hindcasts on both mean and variability studies. Our intention is not to promote the hindcast technique itself since climate hindcast experiment approach has become a widely used method in the Transpose AMIP project and other climate model hindcast studies as we mentioned in the introduction. This is why we submitted this manuscript as a "model experiment description paper". In our earlier hindcast studies (e.g., Xie et al. 2012; Ma et al. 2013 J. Climate), we analyzed the mean biases from two years of short-range hindcasts (May 2008 to April 2010) during Years of Tropical Convection (YOTC). In Ma et al. (2015 JAMES) paper, we proposed a refined hindcast approach in improving the initial atmospheric aerosol profiles and initial land conditions. Based on the refined procedure, we proposed a ''Core'' integration (one-year long) with the refined procedure for a simple, easily repeatable test that allows model developers to rapidly compute appropriate metrics for assessing the impacts of various parameterization changes on the fidelity of cloud-associated processes with available observations. In this manuscript, we applied the refined initialization strategy in Ma et al. (2015 JAMES) and performed this "new" suite of multi-year short-range hindcasts. One can now use this suite of hindcasts to conduct more studies (like the examples listed in this manuscript) which cannot be achieved from single or two years of short-range hindcasts. This is a different purpose from what we proposed in Ma et al. (2015 JAMES). We have strengthened this point in the abstract:*

*"These analyses can only be done through this multi-year hindcast approach to establish robust statistics of the processes under well-controlled large-scale environment because these phenomena are either interannual climate variability or only happen a few times in a given year (e.g. MJO, or cloud regime types)"*

*and in the introduction over Lines 68-73:*

*"This experiment provides an new opportunity to address several modeling issues associated with moist processes, which cannot be achieved from previous short Transpose AMIP II hindcasts (Williams et al. 2013), or one or two years of short-range hindcasts that we conducted in the past (Xie et al. 2012, Ma et al, 2013, Ma et al. 2015). This is because these phenomena are either interannual climate variability or only happen a few times in a given year, and thus we need multi-years to robustly quantify the errors associated with these phenomena."*

*For our initial submission, we did not include detailed information on the initialization procedure since this was already described in Ma et al. (2015 JAMES). We did not want to duplicate the information because the initialization technique is not the main focus of the manuscript. In this revised version of the manuscript, we provided more information on how to conduct the multi-year hindcasts including the initialization information. Recently, we made our scripts and hindcast procedure documentation available to the public on GitHub (https://github.com/PCMDI/CAPT). We also include this information in the revised manuscript over Lines 107-109.*

*Regarding the concern of using CESM1/CAM5/CLM4 as our base model, we understand this model is not currently supported by NCAR anymore. However, simulations from this version are still used by many people including students in universities because the performance of this model is still very good. For model development and diagnosis purpose, people still compare model performance between the latest version and older version to understand the impacts of parameterization changes to the older version. For example, CAM5 results were still being mentioned in the recent CESM workshop in June 2020. As we are preparing another suite of multi-year short-range hindcasts with DOE E3SM/EAM v1, we will compare our results from E3SM to CESM1. The E3SM EAM v1 was originally branching from CAM5.3, which has very similar model performance to CAM5. Therefore, the comparison between the two suites of hindcasts will provides us information on the impacts of parameterization and other changes on cloud process performance. Also note that E3SM/EAM v1 has a new set of atmospheric physical parameterizations that are very similar to CAM6, the latest CAM.*

*Finally, this suite of hindcasts was completed back in 2016 and we have been thinking of more ways on how to better utilize these hindcasts other than what we had planned for. Therefore, it took us a while to prepare this manuscript because we want to provide the community more ways on how to use this type of hindcast experiment.*

*For our responses to other comments, please see our point-by-point responses below in bold.*

**3 Major Concerns**

3.1 Uniqueness of manuscript and relevance for GMD

The manuscript is presented as describing "a multi-year hindcast experiment and its experiment procedure" and giving "examples to demonstrate how one can better utilize simulations from this experiment design." Given that this is the authors' explicitly-stated premise for the paper, it is not immediately clear to me how this manuscript represents a unique contribution to the literature in general, and why it is a good fit for GMD in particular. The authors have written

numerous papers about ensembles of short-duration hindcasts for over 15 years, and in that time they have already made a compelling argument that this experimental design is valuable. Why is yet-another-paper demonstrating the utility of this type of experimental design needed in the literature in general? After reading the manuscript several times, it is still not clear how this manuscript is unique relative to existing literature.

*Response to reviewer:*
***Please see our response earlier regarding the concern of uniqueness of manuscript and relevance for GMD.***

With respect to GMD in particular, there are three types of manuscript (wrt https://www. geoscientific-model-development.net/about/manuscript_types.html) that this could plausibly fit as:

1. Methods for assessment of models,
2. Model experiment description papers, or 3. Model evaluation papers.

If it is a manuscript of type #1, I believe the main contribution here would be "novel ways of comparing model results with observational data." The authors have written extensively about comparing hindcasts with ARM-SGP data (this was one of the earliest uses of CAPT, as far as I can tell), so the 1st example analyzed doesn't seem sufficiently novel. The authors' composite analysis of MJO in hindcasts does seem unique in the literature, so this could considered a novel way of comparing models and observations. However, I would expect this analysis to be emphasized much more prominently in the paper if it was considered the main, novel contribution of the paper; instead, it is one of three examples that the authors run through, and the authors only devote three paragraphs to the topic. Finally, the analysis of diabatic processes associated with ENSO could be unique, but it wasn't clear to me what the authors were trying to show with this analysis, aside from showing that the correspondence between errors in short- and long-duration simulations is robust with respect to ENSO phase.

*Response to reviewer:*
***This manuscript is not under the methods for assessment of models.***

The way the manuscript is written, it seems that the authors are targeting manuscript type #2 "Model experiment description papers", in which case I would expect that the main contribution of the paper is "descriptions of standard experiments for a particular type of model," and "discussion of why particular choices were made in the experiment design and sample model output". The manuscript instead seems to describe a rather vague experimental protocol, consisting of short-duration hindcasts paired with AMIP simulations. The authors also describe their specific implementation of these hindcasts. It also isn't clear how the experimental design presented here differs substantially from other, seemingly similar experimental protocols described in the literature: e.g., Transpose-AMIP II (Williams et al., 2013), and the hindcast approach described by Ma et al. (2015). If it does differ, the authors should explicitly compare and contrast their proposed experimental protocol with those already described in the literature. In particular, Ma et al. (2015)–which this manuscript indicates the hindcast procedure is based on–states that "We also hope to provide guidance for those performing transpose-AMIP/CAPT

simulations with their own climate models for model development or error diagnosis purposes."
That sounds strikingly similar to a "model experiment description paper," which again raises the
unanswered question of how this manuscript represents a unique contribution to the literature.
The authors should also follow the GMD guidelines for this particular manuscript type, including
giving the experimental protocol a name and version number in the manuscript title.

*Response to reviewer:*
*As we responded earlier, we submitted this manuscript as a "model experiment description paper". We aren't sure if this information was directly provided to the reviewers or not. It is shown on the GMD manuscript page ([https://www.geosci-model-dev-discuss.net/gmd-2020-39/#discussion](https://www.geosci-model-dev-discuss.net/gmd-2020-39/#discussion)). As we have explained the uniqueness of this suite of multi-year hindcast experiment and why this is different from we have published in earlier studies including Ma et al. (2015 JAMES), please refer to the response above.*

*From the GMD Manuscript types ([https://www.geoscientific-model-development.net/about/manuscript_types.html#item4](https://www.geoscientific-model-development.net/about/manuscript_types.html#item4)):*

*"For model experiment description papers, similar version control criteria apply as to model description papers: the experiment protocol should be given a version number; a data availability paragraph must be included in the manuscript; boundary conditions should be given a version number and uploaded or made otherwise available; a data availability paragraph must be included in the manuscript; and links to the GMD paper should be included on the experiment website. Since the primary purpose of these papers is to make experiments accessible to the community, all input data required to perform the experiments must be made publicly available"*

*With the description above, it is not clear whether the version number should be on the manuscript title. Nevertheless, we added CESM1 to the manuscript title. The title is now "A multi-year short-range hindcast experiment with CESM1 for evaluating climate model moist processes from diurnal to interannual time scales". We also provided the model version number (CESM1_0_5) and experiment configuration (FC5) in Section 2.1. We added more information regarding how initial conditions were generated in the revised manuscript. The information of boundary conditions and other code information are described in Code and data availability section after the Summary.*

In its current form, I don't think this manuscript is appropriate to be considered a paper of type
**3. The authors' analysis focuses on an outdated version of CESM that is no longer supported**
and is decreasingly used. Further, the current manuscript doesn't refer to any of the (extensive)
existing literature evaluating CESM1, so there is currently no way to determine how/why this
paper is unique relative to other model evaluation papers. (All of that said, I strongly suspect the
authors weren't targeting this manuscript type, in which case such a literature review of CESM1
would not be needed.)

*Response to reviewer:*
*This manuscript is not a model evaluation type paper.*

Williams, K.D., A. Bodas-Salcedo, M. Déqué, S. Fermepin, B. Medeiros, M. Watanabe, C. Jakob, S.A. Klein, C.A. Senior, and D.L. Williamson, 2013: The Transpose-AMIP II Experiment and Its Application to the Understanding of Southern Ocean Cloud Bi- ases in Climate Models. J. Climate, 26, 3258–3274, https://doi.org/10.1175/JCLI-D- 12-00429.1

Ma, H.-Y., C. C. Chuang, S. A. Klein, M.-H. Lo, Y. Zhang, S. Xie, X. Zheng, P.-L. Ma, Y. Zhang, and T. J. Phillips (2015), An improved hindcast approach for evaluation and diagnosis of physical processes in global climate models, J. Adv. Model. Earth Syst., 7, 1810–1827, doi:10.1002/ 2015MS000490.

3.2 Repeatability of experiment

Assuming that the authors are targeting this manuscript as a "Model experiment description paper", the manuscript does not appear to provide adequate detail for an external group to be able to replicate this experimental protocol a constrained way. More detail would need to be provided for the initial condition generation procedure, the nudging procedure, for the land-surface spinup procedure, and greenhouse gas and solar forcing (do these follow the AMIP protocol?). It also seems like it would be useful to guide other modeling groups on what variables should be saved as output: otherwise, how would experiments be intercompared?

*Response to reviewer:*
*For our initial submission, we did not include detailed information on the initialization procedure since this was already described in Ma et al. (2015 JAMES). We did not want to duplicate the information because the initialization technique is not the main focus of the manuscript and modeling groups who have the capability to conduct hindcast studies already have their own initialization strategy. Nevertheless, in the revised manuscript, we provided more information on how to conduct the multi-year hindcasts including the initialization generation procedure, the nudging procedure, land-surface spin-up procedure. The greenhouse gas and solar forcing is based on the setup of the CESM1 FC5 compset, which corresponds to the year 2000 level for all the simulation period. This is because the CMIP5 forcing data does not go beyond 2005. We also don't want the interannual variations in the solar and greenhouse gas forcing to affect our results so that we can better identify possible causes of model biases associated with model parameterizations. Recently, we made our scripts and hindcast procedure documentation available to the public on GitHub (https://github.com/PCMDI/CAPT). We also include this information in the revised manuscript over Lines 107-109.*

*Regarding what variables should be saved in the experiment, this really depends on the subjects of the studies. For example, we participated in the MJO diabatic heating process model intercomparison project and we also organized the US summertime warm bias model intercomparison project. There is an overlap of output variables but there are also variables, which are unique to MJO studies, such as the diabatic heating profiles and all the state variable budget terms. While in the warm bias study, we requested more variables associated with clouds and surface processes. Nevertheless, we do have a list of output variable available to others over [https://portal.nersc.gov/archive/home/h/hyma/www/CAPT/CAPT_Long/CAPT_Long_output_cesm1_0_5_v5.pdf](https://portal.nersc.gov/archive/home/h/hyma/www/CAPT/CAPT_Long/CAPT_Long_output_cesm1_0_5_v5.pdf). This information is under the Code and data availability section.*

Specifically, a number of questions come to mind that could impact the results from other modeling groups implementing this experiment:

- What method(s) should be used to horizontally and vertically remap the ERA- Interim data onto the model grid?

  *Response to reviewer:*
  *The goal is to preserve the state variable information from the reanalysis as much as possible after the interpolation. In the present study, we applied the bilinear interpolation with ESMF regridding utility for the horizontal remap for all the state variables. For vertical remap, we follow the procedure used at ECMWF when initializing model with foreign analysis: Quadratic interpolation is used for temperature remap, linear interpolation is used for specific humidity, and a combination of linear and quadratic interpolation is used for zonal and meridional winds. The vertical interpolation scripts were originally constructed following the ECMWF procedure by one of our colleagues who has retired back in 2013. We didn't have the reference listed here because we can't find the link to the reference anymore after an update of ECMWF webpage.*

  *Note that based on the experience from the past model intercomparison projects that we have organized or participated in, there were generally no guidelines on what method(s) should be used to do horizontal or vertical remap. This is because each model group, especially for operational centers, already have a set of tools to do this.*

- Do any adjustments need to be made to the initial conditions to avoid spurious gravity waves associated with differences in topography between ERA-Interim and the given model?

  *Response to reviewer:*
  *We applied a spatial smoothing for the state variables. This is based on Gerrity and McPherson (1970).*

  *Gerrity, J. P. and R. D. McPherson, 1970: Noise analysis of a limited-area fine-mesh prediction model. ESSA Technical Memoranda, WBTM NMC 46, PB-191-188. 81pp.*

  *We also adjusted the surface pressure associated with differences in topography between ERA-Interim and CAM5 using hydrostatic approximation.*

  *Note that the above information is available in the documentation over our GitHub repository (https://github.com/PCMDI/CAPT).*

- When running 'nudging' simulations to generate non-state-variable ICs:
  - – What nudging method is used (this is provided in Ma et al. (2015), but it should be included here for completeness)?

    *Response to reviewer:*

*A U, V only nudging was used in the nudging run. We follow the recommendation from Zhang et al. (2014 ACP) and also made a few test runs to reach the same conclusion in Ma et al. (2015 JAMES). This information is included in the revised manuscript over Lines 98-101.*

*Reference:*
*Zhang, K., Wan, H., Liu, X., Ghan, S. J., Kooperman., G. J., Ma, P.-L., Rasch, P. J., Neubauer, D., and Lohmann, U.: Technical note: On the use of nudging for aerosol–climate model intercomparison studies, Atmos. Chem. Phys., 14, 8631–8645, 2014.*
*Ma, H.-Y., C. C. Chuang, S. A. Klein, M.-H. Lo, Y. Zhang, S. Xie, X. Zheng, P.-L. Ma, Y. Zhang, and T. J. Phillips (2015), An improved hindcast approach for evaluation and diagnosis of physical processes in global climate models, J. Adv. Model. Earth Syst., 7, 1810–1827, doi:10.1002/ 2015MS000490.*

- – How are the ERA-Interim data interpolated from the ERA-Interim times to the model's current time (nearest-neighbor, linear, spline, other?)

*Response to reviewer:*
*In the nudging simulation, the reanalysis data are linear interpolated between two time steps. Note that the nudging and interpolation codes were created by a NCAR software engineer and we didn't make any more changes.*

- – Is the nudging simulation run in an identical configuration to the AMIP run, with the exception of the nudging term?

*Response to reviewer:*
*That is correct. We also added this information in the revised manuscript over Lines 116-117.*

- – Why aren't the land-surface conditions from the nudging simulation used for the land-surface initial condition in the hindcasts?

*Response to reviewer:*
*This is because in a nudging simulation, biased precipitation, winds, and surface fluxes are allowed to pass to the land model. This will cause larger biases in the simulated soil moisture and temperature. This was discussed in Ma et al. (2015 JAMES). Therefore, we did not mention this in the present manuscript since the focus is on the multi-year hindcast experiment, not the initialization procedure.*

- When running the offline land-surface spinup simulation:
  - – How are surface enthalpy and moisture fluxes calculated? (readers shouldn't have to read through old CESM documentation to figure out what the CESM-community-specific lingo for 'offline' land surface simulation refers to)

    *Response to Reviewer:*

*These are two different questions regarding what is an offline simulation, and how are surface enthalpy and moisture fluxes calculated in CLM.*

*For an offline land model simulation, we stated in Section 2.1 that it is a land model simulation forced by reanalysis and observations including precipitation, surface winds, and surface radiative fluxes. We added another sentence after the original sentence to make this clear. This is now "Land initial conditions are taken from an offline land model simulation (I2000 compset) forced by reanalysis and observations including precipitation, surface winds, and surface radiative fluxes (N. Viovy 2013, unpublished data) rather than coupled it to an active atmospheric model" over Lines 102-104.*

*Regarding the calculation of the surface enthalpy and moisture fluxes, this is really not the focus of this manuscript. For us (atmospheric scientists), we also need to read through the CESM documentation to know the exact equations to calculate these fluxes since we are not land model developers. In short, the surface fluxes are based on Monin-Obukhov Similarity Theory. Without copying the entire chapter from the CLM4 documentation to here, we refer the reviewer to see Section 5 Momentum, Sensible Heat, and Latent Heat Fluxes of the CLM4 technical note on how to calculate these fluxes.*

*[http://www.cesm.ucar.edu/models/ccsm4.0/clm/CLM4_Tech_Note.pdf](http://www.cesm.ucar.edu/models/ccsm4.0/clm/CLM4_Tech_Note.pdf)*

- – How should chemical fluxes be handled if needed by the land-surface model (e.g., C/N in the case of CLM5 in prognostic Carbon/Nitrogen mode)?

  *Response to Reviewer:*
  *This is an open question since we do not turn on the C/N option in all of our hindcasts. One could perform the same offline land model simulation with C/N turned on although the spin-up time may require much longer time (more cycles).*

- – Why is the N. Viovy dataset used for forcing the offline land-surface spinup simulation?

  *Response to Reviewer:*
  *We selected this option (CRUNCEP) in CLM because they offer the most recent years of atmospheric forcing data at that time. Also, the land state from offline simulation is superior than that from a nudging simulation (Ma et al. 2015 JAMES).*

- – How are data from these observational datasets interpolated to the model grid (nearest neighbor, linear, conservative remapping, other)?

  *Response to Reviewer:*

> *We used the default option in the model setup when running the offline land model simulation. The default method is bilinear interpolation according to CESM1.0 documentation:*

- The authors should provide the code that they use to generate the initial conditions, since there are undoubtedly numerous other questions about implementation of this experiment that would arise when external groups attempt to implement this protocol.

> *Response to Reviewer:*
> *We have now made our scripts and hindcast procedure documentation available to the public on GitHub (https://github.com/PCMDI/CAPT). We also included this information in the revised manuscript over Lines 107-109.*

3.3 Old model version

The authors state that "model developers can achieve additional useful understanding of the underlying problems in model physics by conducting a multi-year hindcast experiment." However, this statement is undermined by the author's use of CESM 1.0.5, which stopped being supported and developed by NCAR years ago. The authors state "Although newer version [sic] of the CAM and CLM is now available (CAM6/CLM5), similar systematic errors associated with moist processes remain present in the latest model version," but they state this without any reference to manuscripts that support this statement. Further, CESM2 contains numerous upgrades to key parameterizations associated with moist processes: in particular the adoption of CLUBB, MG2 prognostic microphysics, and a retuning of the convection parameterization to "increase the sensitivity to convection inhibition". Because of all these changes, I don't see how the results from this dataset could be used to inform model development in CESM2, which is the only version of CESM under active development.

If this is indeed intended to be a "Model experiment description paper," then this point is somewhat less relevant. However, the authors should be more forthcoming about these caveats and the utility of the dataset produced as part of this paper. This paper would also be much more impactful if the authors made some specific comments about what (if anything) would need to be done to implement this experimental design for CESM2.

*Response to Reviewer:*
*This manuscript is indeed a "Model experiment description paper". We have provided our response to this concern in the beginning of our responses.*

**4 Specific, minor issues**

pg 1, line 21: "associated parameterized" –> "associated with parameterized"
*Response to reviewer:*
*Revised.*

pg 3, line 79: "Section 4 present" –> "Section 4 presents"
*Response to reviewer:*
*Revised.*

pg 4, line 104: "output at model timestep" –> "output for every model timestep" (?)
*Response to reviewer:*
*Revised*

pg 5, lines 156-158: this is one of several theories for the MJO (e.g., see Yang and Ingersoll, 2011), so these feedback processes may not all be necessary
*Response to reviewer:*
*We revised the sentence to "These feedback processes may contribute to better MJO simulations if they are well represented in the GCMs", so the sentence does not seem to suggest that these feedback processes are all necessary for realistic MJO simulations.*

pg 6, line 165: what is Q1?
*Response to reviewer:*
*We added the following short paragraph for the information of Q1 in the revised manuscript:*

*"The diabatic heating rate or apparent heating of large-scale motion system (Q1) consists of the heating due to radiation, the release of latent heat by net condensation, and vertical convergence of the vertical eddy transport of sensible heat (Yanai et al. 1973). In CESM1/CAM5, Q1 can be calculated through summing up all the tendency terms with all the diabatic processes."*

*Reference:*
*Yanai, M., Esbensen, S., and Chu, J.–H.: Determination of bulk properties of tropical cloud clusters from large-scale heat and moisture budgets, J. Atmos. Sci., 30, 611–627, 1973.*

pg 7, line 220: "the response SST anomalies is much superior" –> "the response to SST anomalies is much superior"
*Response to reviewer:*
*We revised the sentence to "the response of those fields to SST anomalies in Figure 7".*

pg 7, line 221: "the result of poor circulation" –> "the result of errors in circulation" ('poor circulation' is usually reserved for describing anatomical difficulties with blood flow)
*Response to reviewer:*
*Revised as suggested.*

pg 8, line 253: "the annually cloud error metrics" –> "the annually-averaged cloud error metrics" (?)
*Response to reviewer:*
*Revised as suggested.*

pg 9, lines 276-277: "These comparisons identify. . . " this statement only makes sense to include if the authors repeat the experiments with an actively-developed model version.

*Response to reviewer:*

*Yes, we do plan to perform multi-year hindcasts with an actively-developed model version (E3SM) and compare with the CESM1-CAM5 results. We added "We will also compare the results from E3SM to CESM1 to understand the impact of parameterization and model changes to the performance of moist processes since the atmospheric component of E3SM was originally branching form CAM5.3, which has very similar performance as CAM5 (Xie et al. 2018, Rasch et al. 2019). Note that E3SM version 1 has a new set of atmospheric physical parameterizations that are very similar to CAM6, the latest CAM." in the revised manuscript over Lines 353-356.*

Yang, D. and A.P. Ingersoll, 2011: Testing the Hypothesis that the MJO is a Mixed Rossby–Gravity Wave Packet. J. Atmos. Sci., 68, 226–239, https://doi.org/10.1175/2010JAS3563.1

---

## Author Comment (AC3) · 9 Jul 2020

**General comments:**

The authors describe a multi-year hindcast experiment, in which short (3-day) initialised forecasts are performed each day with a climate model. The aim of the experiment is to understand the systematic errors associated with moist processes, which typically appear within the first few days of a model simulation. By performing short forecasts daily for 16 years, the authors are able to generate robust statistics of these systematic errors. This allows them to understand whether the errors depend on the phase of natural variability, or to aggregate statistics across all forecasts to minimise the effects of that natural variability. The authors provide three examples of the types of analysis that could be performed with this hindcast set: the diurnal cycle of clouds over the central United States, the propagation of the Madden-Julian Oscillation through the Maritime Continent and the response of tropical precipitation and circulation to the El Nino-Southern Oscillation.

While I believe that the multi-year hindcast experiment is a useful framework to understand model systematic errors, it is not clear that this manuscript advances either the experiment protocol itself or the methods used to analyse those experiments. The authors have performed and analysed similar experiments in the past, as have other groups. The analysis and conclusions presented here are brief, mainly descriptive, and occasionally erroneous. The authors do not consistently show how the errors diagnosed in the short-range hindcasts differ from those diagnosed from AMIP experiments, which is important to demonstrate the value of their framework and to understand the AMIP errors. The interpretation of MJO propagation in such short simulations is also problematic. Finally, the authors motivate their experiment framework by invoking process-level improvements in models, but it is not clear how the analysis shown here would directly inform specific, targeted model development efforts to reduce these biases. I expand on these points below.

**Response to reviewer:**

**We thank the reviewer for all the comments. Those certainly helped to improve our manuscript.**

Regarding the new uniqueness of the current experimental design, this is our first time documenting this suite of multi-years (1997-2012, 16 years) hindcasts and proposing on how one can better utilizing these hindcasts on both mean and variability studies. Our intention is not to promote the hindcast technique itself since climate hindcast experiment approach has become a widely used method in the Transpose AMIP project and other climate model hindcast studies as we mentioned in the introduction. This is why we submitted this manuscript as a "model experiment description paper". In our earlier hindcast studies (e.g., Xie et al. 2012; Ma et al. 2013 J. Climate), we analyzed the mean biases from two years of short-range hindcasts (May 2008 to April 2010) during Years of Tropical Convection (YOTC). In Ma et al. (2015 JAMES) paper, we proposed a refined hindcast approach in improving the initial atmospheric aerosol profiles and initial land conditions. Based on the refined procedure, we proposed a "Core" integration (one-year long) with the refined procedure for a simple, easily repeatable test that allows model developers to rapidly compute appropriate metrics for assessing the impacts of various parameterization changes on the fidelity of cloud-associated processes with available observations. In this manuscript, we applied the refined initialization strategy in Ma et al. (2015 JAMES) and performed this "new" suite of multi-year short-range hindcasts. One can now use this suite of hindcasts to conduct more studies (like the examples listed in this manuscript) which cannot be achieved from single or two years of short-range hindcasts. This is a different purpose from what we proposed in Ma et al. (2015 JAMES). We have strengthened this point in the abstract:

"These analyses can only be done through this multi-year hindcast approach to establish robust statistics of the processes under well-controlled large-scale environment because these phenomena are either interannual climate variability or only happen a few times in a given year (e.g. MJO, or cloud regime types)"

**and in the introduction over Lines 68-73:**

"This experiment provides an new opportunity to address several modeling issues associated with moist processes, which cannot be achieved from previous short Transpose AMIP II hindcasts (Williams et al. 2013), or one or two years of short-range hindcasts that we conducted in the past (Xie et al. 2012, Ma et al, 2013, Ma et al. 2015). This is because these phenomena are either interannual climate variability or only happen a few times in a given year, and thus we need multi-years to robustly quantify the errors associated with these phenomena."

Please see below our point-by-point responses to your comments.

Major comments:

1. This manuscript is a GMD "model experiment description paper", which are "descriptions of standard experiments for a particular type of model". This manuscript is missing many of the GMD criteria for such a paper, including a name for the experiment, a version number for the protocol, and a version number for the boundary and initial conditions. The description of the experiment design in the paper also falls short of expectations; it lacks much of the detail that one would find in the description for a MIP, for instance, which seems to be the GMD gold standard. In particular, the following details are not clear:

**Response to reviewer:**

We submitted this manuscript as a "model experiment description paper". We did not submit this as a "MIP" type protocol although we plan to use to this experiment design as one of the experiment categories in the Diurnal Cycle of Precipitation (DCP, https://portal.nersc.gov/cfs/capt/diurnal/) model intercomparison project under the Global Energy and Water cycle Exchanges (GEWEX) Global Atmospheric System Studies (GASS).

From the GMD Manuscript types (https://www.geoscientific-model- development.net/about/manuscript\_types.html#item4):

"For model experiment description papers, similar version control criteria apply as to model description papers: the experiment protocol should be given a version number; a data availability paragraph must be included in the manuscript; boundary conditions should be given

a version number and uploaded or made otherwise available; a data availability paragraph must be included in the manuscript; and links to the GMD paper should be included on the experiment website. Since the primary purpose of these papers is to make experiments accessible to the community, all input data required to perform the experiments must be made publicly available"

We added CESM1 to the revised manuscript title "A multi-year short-range hindcast experiment with CESM1 for evaluating climate model moist processes from diurnal to interannual time scales". We provided the model version number (CESM1\_0\_5) and experiment configuration (FC5) in Section 2.1. We added more information regarding how initial conditions were generated in the revised manuscript. The information of boundary conditions and other code information are already described in Code and data availability section after the Summary. We also searched similar model experiment description papers published on GMD webpage. Here is a list of a few manuscript links. We also followed how they provided their model and experiment information.

https://gmd.copernicus.org/articles/11/3865/2018/ https://gmd.copernicus.org/articles/10/2833/2017/gmd-10-2833-2017.pdf https://gmd.copernicus.org/articles/10/1665/2017/gmd-10-1665-2017.pdf

a. Do all initial conditions come from the nudged simulations, or do some fields come directly from ERA-Interim? The current description at L86 is not clear on this.

Response to reviewer:

In Section 2.1, we did state in our initial submission from L86 that we applied the "horizontal velocities", "temperature", "specific humidity" and "surface pressure" from the ERA-Interim Reanalysis (Dee et al. 2011) for the initial atmospheric states. A nudging simulation with CAM5/CLM4 was also performed to acquire "other necessary variables" (e.g., cloud and aerosol fields), which are not available from the ERA-Interim Reanalysis for the atmospheric initial conditions. It will be helpful if the reviewer can explicitly point out which part is not clear.

b. What fields are nudged in the nudging simulations?

Response to reviewer:

We now included more information about this in the revised manuscript over Lines 98-100: "A nudging simulation (horizontal velocities nudging only following Zhang et al. 2014) with CAM5/CLM4 was also performed to acquire other necessary variables (e.g., cloud and aerosol fields)".

c. What reanalysis and observation datasets were used force the offline land simulation? *Response to reviewer:*

The reanalysis and observation datasets used to force the offline land simulation is the CRUNCEP dataset. It is one of the options in driving the offline CLM simulations. It is provided by NCAR. The reason we chose it is because the forcing data covers the most recent years. This dataset is widely used in the CLM community. Unfortunately, we cannot find a reference for this dataset. In the revised manuscript, we revised the sentence to "Land initial conditions are taken from an offline land model simulation (I2000 compset) forced by

reanalysis and observations including precipitation, surface winds, and surface radiative fluxes (CRUNCEP, N. Viovy 2013, unpublished data) rather than coupled it to an active atmospheric model" over Lines 102-104 in Section 2.1.

d. Is land initialised to a climatology from the offline simulations, or to the value on a particular day? If the latter, is the value taken from the last spinup cycle of the land model? *Response to reviewer:*

The land was initialized to the values on a particular calendar day from the offline simulations. We further clarified this in the revised manuscript "The offline land model simulation started from 1990 to 2012 and we performed five cycles (1990 to 2012) for the offline simulation to allow proper spin-up of the land conditions. After that, we continued the offline land model simulation to the desire starting date and use the land model restart file (.r file) as the land initial condition" over Lines 104-107.

e. Do greenhouse gases and aerosols follow the AMIP specification? *Response to reviewer:*

The model configuration of greenhouse gases and aerosols is the same for the nudging, hindcast and AMIP experiments. In the revised manuscript, we revised the sentence to "We also conducted a 16-year long AMIP simulation with the same model for the same period. In this AMIP simulation, the state of the atmosphere evolves freely without constraints. Note that the nudging simulation mentioned above has the same model configuration as the AMIP simulation with the exception of the nudging terms" over Lines 115-117.

Also, I note that the offline land simulation data are not published, according to the citation in the manuscript, but GMD specifies that all boundary conditions must be published under version control, as part of the paper.

**Response to reviewer:**

As we stated earlier, the reanalysis and observation datasets used to force the offline land simulation is the CRUNCEP dataset. It is one of the options in driving the offline CLM simulations. It is provided by NCAR and the reason we chose it is because the forcing data covers to the most recent years. This dataset is widely used in the CLM community. Unfortunately, we cannot find a refence for this dataset.

2. The novel contribution of the manuscript to the design and analysis of short-range hindcast experiments is not clear. The authors have conducted similar experiments in the past through the CAPT project (e.g., Ma et al., 2015); other similar experiment designs exist, such as Transpose-AMIP. This is far from the first paper to propose such an approach, which I think the authors would acknowledge. What is new and innovative about this experiment design? How does this experiment design enable new understanding of the development of systematic errors related to moist processes, beyond that which could be achieved through existing protocols? *Response to reviewer:*

Regarding the new uniqueness of the current experiment, this is our first time documenting this suite of multi-years (1997-2012, 16 years) hindcasts and proposing on how one can better utilizing these hindcasts on both mean and variability studies. Our intention is not to promote the hindcast technique itself since climate hindcast experiment approach has become a widely used method in the Transpose AMIP project and other climate model hindcast studies as we

mentioned in the introduction. This is why we submitted this manuscript as a "model experiment description paper". In our earlier hindcast studies (e.g., Xie et al. 2012; Ma et al. 2013 J. Climate), we analyzed the mean biases from two years of short-range hindcasts (May 2008 to April 2010) during Years of Tropical Convection (YOTC). In Ma et al. (2015 JAMES) paper, we proposed a refined hindcast approach in improving the initial atmospheric aerosol profiles and initial land conditions. Based on the refined procedure, we proposed a "Core" integration (one-year long) with the refined procedure for a simple, easily repeatable test that allows model developers to rapidly compute appropriate metrics for assessing the impacts of various parameterization changes on the fidelity of cloud-associated processes with available observations.

In this manuscript, we applied the refined initialization strategy in Ma et al. (2015 JAMES) and performed this "new" suite of multi-year short-range hindcasts. One can now use this suite of hindcasts to conduct more studies and analyses (like the examples listed in this manuscript) which cannot be achieved from single or two years of short-range hindcasts. For example, there won't be enough cases to make robust statistical composites for the cloud regimes over the DOE ARM SGP site with only one single year of short range hindcasts. This is the same for the MJO phase composites. Further, one cannot answer the question of whether systematic errors of moist processes show significant interannual variation in the mean state biases, or the question of whether or not errors in the response of precipitation and surface fluxes to SST anomalies associated with ENSO can be attributed to parameterization errors or whether errors in the circulation response to SST anomalies also contribute, with only single or two years of shortrange hindcasts. All these analyses require a suite of multi-year hindcasts and this is the uniqueness of this suite of multi-year hindcasts.

We have strengthened this point in the abstract:

"These analyses can only be done through this multi-year hindcast approach to establish robust statistics of the processes under well-controlled large-scale environment because these phenomena are either interannual climate variability or only happen a few times in a given year (e.g. MJO, or cloud regime types)"

and in the introduction over Lines 68-73:

"This experiment provides an new opportunity to address several modeling issues associated with moist processes, which cannot be achieved from previous short Transpose AMIP II hindcasts (Williams et al. 2013), or one or two years of short-range hindcasts that we conducted in the past (Xie et al. 2012, Ma et al, 2013, Ma et al. 2015). This is because these phenomena are either interannual climate variability or only happen a few times in a given year, and thus we need multi-years to robustly quantify the errors associated with these phenomena."

3. At various points in the paper (e.g., L129, L155, L277), the authors motivate their analysis of short-range hindcast simulations by invoking parameterisation development or parameterisation evaluation. Yet, it is difficult to see how a parameterisation developer could gain useful insight from the simulations the authors present. If I were developing a convection parameterisation, I do not know how I would be able to apply the authors' analysis to improve that parameterisation.

The authors do not test the sensitivity to the choice of model parameterisation, let alone to choices of particular model parameters. To make such a test would require repeating the full hindcast set, potentially several times to test the sensitivity to various choices. I would expect that this would be costly, both in computational and human resources. Is there actionable information for parameterisation improvement here? Can the authors point to specific parameterisation developments that were made in response to their previous work with short-range hindcasts?

**Response to reviewer:**

The purpose of this is manuscript is a "model experiment description paper", not a model evaluation paper. We only provided examples of possible studies on how to utilize this suite of multi-year hindcasts. We did not provide vigorous evaluation of individual model issues in this manuscript since that is beyond the scope of this manuscript as a model experiment description paper. Nevertheless, some information from the current example does provide insights on parameterization issues. For example, the cloud regime analysis suggests that the model shallow convection scheme is not able to simulate the shallow convection regime well. For afternoon deep convective cloud regime, the model cannot simulate the transition from shallow to deep convective clouds. The deep convection clearly starts too early. These issues all point to specific model parameterizations as we mentioned in the manuscript. The contribution from large-scale state is much smaller because the large-scale state is still very close to the reanalysis at Day 2 hindcasts.

Regarding repeatability of the entire experiment, the first author alone performed another suite of 50-day long hindcasts initialized every day starting at 00Z from 1997 to 2012 with CAM5 within a two-week period. The process can be faster with a better strategy on storing the output because the disk quota was constantly reached. Since each hindcast is independent and can be completed very fast from less than half an hour to two hours depending on the speed of the computing system, one can easily bundle as many hindcasts as possible into one job submission. We now have a section in the revised manuscript to describe this process (Section 2.2 Strategy on performing the multi-year hindcasts on a high performance computing system).

For sensitivity tests to various parameter choices for a specific scheme or parameterization, one does not have to perform this suite of multi-year hindcast experiment. Instead, one can perform "core experiment" (i.e., series of short-range hindcast over one-year period), as we proposed in Ma et al. (2015 JAMES). These two types of experiments: "multi-year hindcast experiment" and "core experiment" have different purposes in studying model biases associated with cloud processes. Further, one can perform a set of hindcasts just for the set of key dates with the phenomena of interest (e.g. days with sallow cumulus at ARM SGP, or phase 3 of various MJOs).

We added this sentence: "For sensitivity tests to various parameter choices for a specific scheme or parameterization, it is not necessary to perform this suite of multi-year hindcast experiment once the issue has been identified. Instead, one could perform a "core experiment" (i.e., series of short-range hindcast over one-year period) as we proposed in Ma et al. (2015), or perform a set of hindcasts just for the set of key dates with the phenomena of interest (e.g.

days with sallow cumulus at ARM SGP, or phase 3 of various MJOs, which we will introduce in the following text)" in the revised manuscript over Lines 148-153 to further clarify this.

Regarding our previous work in guiding parameterization development, we listed many references in the introduction already. For specific example, in Zheng et al. (2017), we identified an issue of a cloudy planetary boundary layer oscillation related to interaction between CLUBB scheme and MG2 microphysics scheme in CAM5. The issue is later fixed in the CAM5/CAM6 and the U.S. DOE E3SM development. There are also many unpublished studies that our hindcast approach was used to guide the development of a scheme or diagnose parameterization errors. Specifically, we used short range hindcasts in testing the candidate convection schemes during the CAM6 and E3SM model development.

4. The authors state that their aim is to understand which of the errors seen in AMIP simulations are due to local-scale errors in moist processes, and which are due to errors in larger-scale or remote processes (e.g., L274-275). However, for the cloud regimes analysis (section 3.1) and the MJO propagation analysis (section 3.2), the authors analyse only the short-range hindcasts, without reference to the AMIP simulation. Thus, it is not possible to understand the relative contributions of moist-process errors. I suggest adding similar analysis for the AMIP simulation, as the authors have done for the ENSO analysis (section 3.3). Otherwise, the value of the short-range hindcasts over the AMIP simulations is not particularly clear.

**Response to reviewer:**

Comparing hindcasts to AMIP simulations to identify errors in parameterizations or largescale sate is certainly beneficial. However, in some targeted studies, such as field campaign or cloud regime/MJO phase in the present manuscript, the comparison to AMIP simulations may not be possible or necessary. For cloud regimes and MJO propagation analyses, the motivation is to better isolate the bias contribution from parameterizations with a wellcontrolled large-scale state. As we stated in the manuscript and our response to comment 3, the analysis of cloud regimes certainly provides information on the issues of convection schemes in the model.

Regarding making the same AMIP composites for the comparison, it is not possible due to the definition of the cloud regimes. For the shallow convection regimes, these days are defined as precipitation rate = 0 mm day-1 at all hours of the day, and shallow cumulus clouds are identified by Berg and Kassianov (2008), who first selected cumulus clouds based on fine temporal resolution ARSCL data at ARM SGP and then manually scrutinized cloud images taken by the Total Sky Imager (available online at http://www.arm.gov/ instruments/tsi) to eliminate low cloud types other than shallow cumulus. For MJO phase composites, one can certainly follow the definition in Wheeler and Hendon (2004). However, the days for MJO phase composites from the AMIP simulation will not correspond to the actual observation days. Also, the large-scale state in the AMIP simulation is not necessarily similar to that in the observations/reanalysis. One can obtain information about the bias correspondence between long-term climate bias and short-term hindcast bias. However, this is not what we intend to show for the MJO phase composite studies.

References:

Berg, L. K., and Kassianoy, E. I.: Temporal variability of fair- weather cumulus statistics at the ACRF SGP site, J. Climate, 21, 3344–3358, 2008. Wheeler, M. C., and Hendon, H. H.: An all-season real-time multivariate MJO index: Development of an index for monitoring and prediction, Mon. Weather Rev., 132, 1917–1932, 2004.

5. The MJO analysis seems incomplete and problematic. In particular:

**Response to reviewer:**

As we stated earlier, the purpose of this is manuscript is a "model experiment description paper", not a model evaluation paper. We only provided examples of possible studies to utilize this suite of multi-year hindcasts. We are not vigorously evaluating individual model issues mentioned in this study.

**Regarding "problematic" issue, please see our response below in 5b.**

a. The authors show diabatic heating (Q1) profiles from the model, but do not compare them against observations (e.g., satellite-derived heating profiles). Yet, they make statements that the model heating is "very weak" (L186) and "not restricted to low levels". This suggests a bias, but the reader cannot judge the bias as there is no truth against which to compare!

Response to reviewer:

Over Line 186. The sentence in the manuscript is:

"During Phases 2 and 3 when the MJO is over the Indian Ocean, the anomalous Q1 profiles reveal that the magnitude of shallow heating is very weak (<0.4 K day-1) to the east over the region of suppressed convection between 100°E and 120°E in Phase 2 and the heating is not restricted to low levels between 120°E and 150°E in Phase 3."

Here, we were just describing the feature from the model simulations.

After this sentence, we stated:

"Instead, there is an anomalous heating associated with deep convection in Phase 3, which is not evident in the observations as indicated from previous studies (e.g., Jiang et al. 2011). This suggests that the model fails to simulate the pre-conditioning moistening processes and the gradual transition from shallow to deep convection as MJO propagates."

We didn't show observed heating profiles here is because the actual bias magnitudes of heating profiles are not the focus. Rather, we wanted to highlight the absence of low-level shallow convection heating in the model by Day 3. The existence of shallow convection ahead of the core of deep convection is a well-known MJO feature (e.g., Johnson et al. 1999; Kikuchi and Takayabu 2004; Chen and Del Genio 2009; Tromeur and Rossow 2010; Powell and Houze 2013; Xu and Rutledge 2014). Instead, the model shows heating profiles associated with deep convection. The absence of low-level shallow convection heating is very obvious, and we think we don't really need observation to prove what has been published in many previous literatures. In the revise manuscript, we further highlight this feature that we want to focus. The revised sentence is: "Instead, there is an anomalous heating associated with deep convection in Phase 3, which is not evident in the observations as indicated from many previous studies (e.g., Figure 5a of Jiang et al. 2011). This suggests that the model fails to simulate the preconditioning moistening processes by shallow convection and the gradual transition from shallow to deep convection as MJO propagates by Day 3".

**Reference:**

Chen, Y. H. and A. D. Del Genio, 2009: Evaluation of tropical cloud regimes in observations and a general circulation model. Climate Dyn., 32, doi:10.1007/S00382-008-0386-6, 355-369. Jiang, X., Waliser, D. E., Olson, W. S., Tao, W.-K., L'Ecuyer, T. S., Shige, S., Li, K.-F., Yung, Y. L., Lang, S., and Takayabu, Y. N.: Vertical diabatic heating structure of the MJO: Intercomparison between recent reanalyses and TRMM estimates, Mon. Wea. Rev., 139, 3208-3223, 2011. Johnson R. H. T. M. Rickenbach, S. A. Rutledge, P. E. Ciesielski, and W. H. Schubert, 1999:

Johnson, R. H., T. M. Rickenbach, S. A. Rutledge, P. E. Ciesielski, and W. H. Schubert, 1999: Trimodal Characteristics of Tropical Convection. J. Clim., 12, 2397-2418.

Kikuchi, K. and Y. N. Takayabu, 2004: The development of organized convection associated with the MJO during TOGA COARE IOP: Trimodal characteristics. Geophys. Res. Lett., 31. Powell, S. W. and R. A. Houze, 2013: The cloud population and onset of the Madden-Julian Oscillation over the Indian Ocean during DYNAMO-AMIE. Journal of Geophysical Research: Atmospheres, 118, 10.1002/2013JD020421, 2013JD020421.

Tromeur, E. and W. B. Rossow, 2010: Interaction of Tropical Deep Convection with the Large-Scale Circulation in the MJO. J. Clim., 23, Doi 10.1175/2009jcli3240.1, 1837-1853. Xu, W. and S. A. Rutledge, 2014: Convective Characteristics of the Madden–Julian Oscillation over the Central Indian Ocean Observed by Shipborne Radar during DYNAMO. J. Atmos. Sci., 71, 10.1175/JAS-D- 13-0372.1, 2859-2877.

b. The authors discuss MJO "propagation" through the Maritime Continent (L195), but in such short (3-day) hindcasts there is no "propagation" as such. The model is constantly reinitialised from the nudged simulations, so no MJO event could ever propagation across the Maritime Continent in a single hindcast. Likewise, the authors discuss a lack of "pre-conditioning" and "gradual transition from shallow to deep convection" in the model (L189). These are concepts often used in free-running climate simulations; the degree to which they apply to such short hindcasts is not clear. There can be no "pre-conditioning" or "gradual" transition in the span of a single hindcast.

**Response to reviewer:**

In our analysis, we were not examining just one single hindcast. As we stated in Section 2.1 of the manuscript, "We concatenated each hindcast from 24-48 (48-72) hours lead time to form a pseudo Day 2 (Day 3) time series of 16-year duration from 1997 to 2012". Therefore, we examined the Day 2 hindcast time series over this 16-year period like an AMIP simulation. Granted, the Day 2 pseudo time series is not a continuous time series like the AMIP simulation. Nevertheless, we are examining the features of MJO phase composites rather than discussing propagation of a single MJO event in a single hindcast. Similarly, the "pre-

**conditioning" and "gradual transition from shallow to deep convection" are discussed in those phase composites rather than in a specific MJO event of a single hindcast.**

c. Related to the above, it may be useful to diagnose how well the MJO propagates in the nudged simulation used as initial conditions. I believe comparisons of the nudged simulation against the hindcasts may provide more insight into how much of the lack of MJO propagation is due to errors in moist processes in the hindcasts.

**Response to reviewer:**

We agree that the nudged simulation will be useful and may provide useful information when compared with the hindcasts. We added this information "The proposed experiment and evaluation method also complement the existing ways of climate model evaluation, such as performing GCM simulations in the AMIP, or nudging mode. Comparison among the multi-year hindcasts, AMIP and nudging simulations may provide more insights into these issues mentioned above" as a recommendation in the revised manuscript in the summary section over Lines 339-341.

Note that this manuscript is a "model experiment description paper", not a model evaluation paper. Therefore, further investigation with the nudging simulation for MJO propagation is beyond the scope of this manuscript.

d. The authors mention errors in "interactions with the diurnal cycle of convection over the Maritime Continent" (L200), related to the MJO propagation errors, but provide no evidence or analysis of these.

**Response to reviewer:**

As we mentioned above, this manuscript is a "model experiment description paper", not a model evaluation paper. Therefore, diagnosing the errors in "interactions with the diurnal cycle of convection over the Maritime Continent" (L200), related to the MJO propagation errors, will require further investigation as a separate study.

6. For the ENSO analysis (section 3.3), it is not clear why the authors analyse the ENSO regressions globally, given that the authors are interested in the fast (1-3 day) response to ENSO SST anomalies. The far-field response (e.g., over the Indian Ocean or the Atlantic) takes a few weeks to develop, at least. Thus, I suspect that the errors in these regions are not errors in the response to Pacific SST anomalies in the ENSO region, but rather are errors in response to the local circulation and SST, which may or be not be directly related to ENSO. Perhaps I am missing something, but the interpretation of the analysis here seems to be less straightforward than the authors suggest.

**Response to reviewer:**

The motivation is to "gain insights into whether or not errors in the response of these fields (precipitation, surface radiative and heat fluxes, as well as zonal wind stress) to SST anomalies can be attributed to parameterization errors or whether errors in the circulation response to SST anomalies also contribute", as we stated in the beginning of Section 3.3.1. In AMIP simulations, it is difficult to disentangle errors in the response of these fields to SST anomalies from large-scale state or model parameterizations due to feedback processes. With the multi-year hindcasts, the large-scale state at Day 2 remains close to the initial state, therefore, errors in the response of these fields to SST anomalies can mainly attribute to

parameterization errors. Remote cloud errors are the response to the local circulation and SST anomalies, and that some of those anomalies may not be driven by ENSO. However, when one compares the result of the multi-year hindcasts to the AMIP run, it is only the circulation anomalies (correlated to ENSO) which may not be reproduced in the AMIP run, so that comparison of the cloud results between the hindcast and AMIP run provides information about the role of the errors in the circulation anomalies.

Also, related to the ENSO section:

a. L210: The authors say that there are statistics in Table 2, but I cannot see any! Where are the pattern statistics that the authors mention?

**Response to reviewer:**

**We apologize for this mistake. The table (now Table 3) in now in the revised manuscript.**

b. At several places, (e.g., L216, L220), the authors state the hindcasts are "superior", or have "better agreement" with observations, when compared to the AMIP simulations. This agreement is not obvious, particularly in such small figures. It needs to be quantified statistically. *Response to reviewer:*

With the Table 3, the overall spatial correlation coefficients and RMSEs are indeed superior in the hindcasts than in the AMIP quantitatively.

7. In Figure 8, why are the OLR pattern correlations much lower than for the other variables? *Response to reviewer:*

We think that the possible reason for lower OLR pattern correlations is because OLR performance is mainly associated with the performance of convection, radiation and cloud micro- and macro-physics schemes in the model. Therefore, errors in all these schemes can all contribute to the errors in the OLR simulations. However, this requires further investigation and this is beyond the scope of the current manuscript.

8. L247: The authors suggest that robust model errors can be identified "from only one year of hindcasts with enough ensemble members." How many is enough? Can the authors estimate the number of ensemble members required from their simulations?

Response to reviewer:

Based on our previous study in Ma et al. (2014, JCLI), we found that ensemble members larger than 15 may be enough for identifying robust model errors associated with cloud processes. We added this information in the revision over Lines 298-300: "one may identify robust model errors in the mean state from only one year of hindcasts with enough ensemble members (with ensemble members greater than 15, Ma et al. 2014)".

Reference:

Ma, H. Y., Xie, S., Klein, S. A., Williams, K. D., Boyle, J. S., Bony, S., Douville, H., Fermepin, S., Medeiros, B., Tyteca, S., and Watanabe, M.: On the correspondence between mean fore-cast errors and climate errors in CMIP5 models, J. Climate, 27, 1781–1798, 2014.

9. L249: "short simulations will be effective at reducing moist process errors" - Simulations alone do not reduce errors! Only dedicated model development efforts can reduce these errors.

How does this framework, and the results the authors show, contribute to this effort? See also comment 1 above.

Response to reviewer:

We believe this was a mistake of using the wrong word in the initial submission. We have changed the word from "reducing" to "identifying".

---

## Author Response (AR2)

**Editor's Comment**

Comments to the Author:
Dear Dr. Ma,

Thank you for addressing many of the reviewers' concerns.

The reviewers have pointed out the importance of clearly explaining how this model experiment is unique compared to previously published work. Unfortunately, if the experiments presented here are not different from Ma et al. (2015), it is difficult to understand how this can be published as Model Experiment
Description paper. In response to Reviewer Report #2, please revise the manuscript to include the details needed for community users to replicate the model experiment, such as the methods described in your responses to the initial review from Reviewer #2. I see that a link is provided for a pdf titled "Documentation for Multi-year (1997-2012) CAPT Hindcast Output", but note that the document is dated 1/11/2016, which I think supports two reviewers' concerns that there is not a clear distinction between
this submitted manuscript and the Ma et al. (2015) study. In your revised version, please consider making the distinction of this model experiment from the previous work clear for reviewers, myself, and the readers.

Also, please respond to concerns mentioned in Reviewer Report #3 regarding important caveats in
interpreting model performance using this hindcast approach.

Many thanks,
Christina

*Response to Editor:*
*We thank the Editor for the helpful comments.*

*Regarding the differences between the present manuscript and the work in Ma et al. (2015 JAMES), the latter proposed a refined hindcast approach in improving the initial atmospheric aerosol profiles*
*and initial land conditions. Based on the refined procedure, we proposed a ''Core'' integration (one-year long) with the refined procedure for a simple, easily repeatable test that allows model developers to rapidly compute appropriate metrics for assessing the impacts of various parameterization changes on the fidelity of cloud-associated processes with available observations. In this manuscript, we applied the refined initialization strategy in Ma et al. (2015 JAMES) and performed this "new" suite of multi-*
*year (16) short-range hindcasts. These two are not the same experiments (one year vs sixteen years). Furthermore, we used ECMWF YOTC analysis as the initial conditions for Ma et al. (2015 JAMES), while we used ERA-Interim reanalysis as the initial conditions for these multi-year short-range hindcasts. Therefore, these two papers described two different sets of experiments for two different scientific topics.*
*We indicated this in the introduction on Lines 66-71:*

*"In the present paper, we present a multi-year short-range hindcast experiment and its experimental design for better evaluating both the mean state and variability of atmospheric moist processes in climate models to facilitate model development using the CESM1 as the base model. This experiment provides an new opportunity to address several modeling issues associated with moist processes, which cannot be achieved from previous short Transpose AMIP II hindcasts (Williams et al. 2013), or one or two years of short-range hindcasts that we conducted in the past (Xie et al. 2012, Ma et al, 2013, Ma et al. 2015)."*

*Regarding the "Documentation for Multi-year (1997-2012) CAPT Hindcast Output", this is a document for the hindcast output variables for these multi-year hindcasts. We first completed the multi-year hindcasts back in Jan 2016, and we have been thinking of more ways on how to better utilize these hindcasts other than what we had planned for. Therefore, it took us a while to prepare this manuscript because we want to provide the community more ways on how to use this type of multi-year hindcast experiment. The simulations for Ma et al. (2015 JAMES) were completed much earlier in 2013, which are not the same as our multi-year hindcasts completed in 2016 for the present study.*

*We also included more details on how to generate initial conditions and to perform the hindcasts in the revised manuscript in Section 2.1. We further indicated the location to all the initial conditions if one plan to perform the multi-year hindcasts with CAM5 in the revised manuscript on Lines 138-139: "The location to obtain all the model output and necessary initial conditions to conduct the multi-year hindcasts are described in the Code and data availability Section.", and on Lines 391-392: "The initial conditions are located at (https://portal.nersc.gov/archive/home/h/hyma/www/CAPT/CAPT_Long/IC/)."*

*A detailed step-by-step documentation and associated codes on how to generate initial conditions and how to perform hindcasts is on our GitHub page (https://github.com/PCMDI/CAPT). This information is on Line2 120-121 in Section 2.1.*

*We also wanted to point out that anyone who wants to perform the hindcasts should already have some knowledge on how to perform CAM5 AMIP simulations or follow the procedure on the CESM website. As we indicated in the revision on Lines 122-123: "The way to conduct a single hindcast is the same as performing an AMIP simulation except we use the initial conditions from the procedure described above". It is beyond the scope of this manuscript to describe on how to perform CESM AMIP simulations.*

*For the comments from Reviewer #2 and #3, please see our responses below.*

**Anonymous Referee #2**

I have to say that I am torn about how to approach reviewing the authors' revised manuscript and associated comments.

On one hand, I do think there is value in publishing a paper that describes the numerous details involved in carrying out such a hindcast experiment; and it is a worthwhile contribution that the authors have released the source code for this purpose. I would ultimately like to see this paper published.

On the other hand, I cannot immediately see a path toward revising this paper such that it is suitable for publication: it appears to be fundamentally too duplicative of work already in the literature. Both Reviewer #3 and I raised a major concern about the uniqness of this manscript. The authors argue that this experiment is unqiue relative to the previous literature--and in particular relative to Ma et al. (2015)--because it describes a multi-year hindcast. That is a fair argument. However, Reviewer #3 and I both also raised the concern that the authors did not provide a variety specific details that would be necessary for another modeling group to replicate their experimental design. In this case, the authors' response was to argue that "We did not want to duplicate the information because the initialization technique is not the main focus of the manuscript and modeling groups who have the capability to conduct hindcast studies already have their own initialization strategy." (The authors did provide some--though not all--of these details in the revised manuscript.)

In short, it sounds like the experimental design described in this manuscript is simply the Ma et al. (2015) method, except that it is run for multiple years. Is the addition of running for multiple years sufficently unique to warrant publication of a new experimental design? Perhaps, though that argument is weakend by the existence of other multi-year hindcast methods described in the literature.

I would also like to note that the authors appear to have misunderstood my intention in the section of my previous review where I stated "Specifically, a number of questions come to mind that could impact the results from other modeling groups implementing this experiment." I raised these questions with the intent of suggesting that the manuscript should be revised to address the questions. There were a number of instances where the authors instead directly responded to me without actually modifying the manuscript. These were collectively part of my major concern that the manuscript did not provide sufficient technical detail for a "model experiment description paper," and I

would suggest that the authors revise the manuscript accordingly.

In summary, I cannot recommend this manuscript for publication at this time. I
am going to recommend that it again be returned for major revisions, in hopes
that the authors can provide a revised manuscript that makes a much more compelling and
impactful model experiment description.

*Response to reviewer:*
*We thank the reviewer for all the comments. Those certainly helped to improve our manuscript. We understand that (1) a more detailed hindcast approach description and (2) the uniqueness of the manuscript and too duplicative of work already in the literature are the major concerns for the reviewer.*

*(1) Regarding the details of the hindcast approach, we apologize that we misunderstood the reviewer's intention and did not include all our responses in the last revision of the manuscript. In this revision, we now included all the responses to all the comments relevant to hindcast technique in the section2 in revised manuscript.*

*The following are the sentences that we added in the revised manuscript:*

*Lines 99-105: "We applied the bilinear interpolation for the horizontal remapping for all the state variables to the model grid. For vertical remapping, we follow the procedure used at ECMWF when initializing model with foreign analysis: Quadratic interpolation is used for temperature, linear interpolation is used for specific humidity, and a combination of linear and quadratic interpolation is used for zonal and meridional winds. To avoid spurious gravity waves associated with differences in topography between ERA-Interim and CAM5, we applied a spatial smoothing for the state variables (Gerrity and McPherson 1970). We also adjusted the surface pressure associated with differences in topography between ERA-Interim and CAM5 using hydrostatic approximation."*

*Lines 108-109: "During the nudging simulation, the reanalysis data are linear interpolated between two time steps to match model's current time."*

*Lines 111-113: "We do not use land-surface conditions from the nudging simulation for the land-surface initial condition in the hindcasts. This is because in a nudging simulation, biased precipitation, winds, and surface fluxes are allowed to pass to the land model, which will cause larger biases in the simulated soil moisture and temperature (Ma et al. 2015)."*

*Line 116: "The default bilinear interpolation method is used to interpolate the forcing datasets to the CLM grid."*

*We further indicated the location to all the initial conditions if one plan to perform the multi-year hindcasts with CAM5 in the revised manuscript on Lines 138-139: "The location to obtain all the model output and necessary initial conditions to conduct the multi-year hindcasts are described in the Code*

*and data availability Section.", and on Lines 391-392: "The initial conditions are located at* (https://portal.nersc.gov/archive/home/h/hyma/www/CAPT/CAPT_Long/IC/)."

*A detailed step-by-step documentation and associated codes on how to generate initial conditions and how to perform hindcasts is on our GitHub page (https://github.com/PCMDI/CAPT). This information is on Lines 120-121 in Section 2.1.*

*We also wanted to point out that anyone who wants to perform the hindcasts should already have some knowledge on how to perform CAM5 AMIP simulations or follow the procedure on the CESM website. As we indicated in the revision on Lines 122-123: "The way to conduct a single hindcast is the same as performing an AMIP simulation except we use the initial conditions from the procedure described above". It is beyond the scope of this manuscript to describe on how to perform CESM AMIP simulations.*

*(2) Regarding the comment about the uniqueness of the manuscript and too duplicative of work already in the literature, we take this comment very seriously and want to better address this issue. As we acknowledged in the manuscript, the hindcast technique is the same as Ma et al. (2015). However, our focus is on how we can use this set of multi-year hindcasts to explore science questions that we cannot address from the experiments in our previous studies, such as those experiments conducted in Ma et al. (2015). We indicated this in the introduction on Lines 66-71:*

*"In the present paper, we present a multi-year short-range hindcast experiment and its experimental design for better evaluating both the mean state and variability of atmospheric moist processes in climate models to facilitate model development using the CESM1 as the base model. This experiment provides an new opportunity to address several modeling issues associated with moist processes, which cannot be achieved from previous short Transpose AMIP II hindcasts (Williams et al. 2013), or one or two years of short-range hindcasts that we conducted in the past (Xie et al. 2012, Ma et al, 2013, Ma et al. 2015)."*

*Specifically, in Ma et al. (2015) we proposed a refined hindcast approach in improving the initial atmospheric aerosol profiles and initial land conditions. Based on the refined procedure, we proposed a ''Core'' integration (one-year long) with the refined procedure for a simple, easily repeatable test that allows model developers to rapidly compute appropriate metrics for assessing the impacts of various parameterization changes on the fidelity of cloud-associated processes with available observations. In this manuscript, we applied the refined initialization strategy in Ma et al. (2015 JAMES) and performed this "new" suite of multi-year (16) short-range hindcasts. These two are not the same experiments (one year vs sixteen years). Furthermore, we used ECMWF YOTC analysis as our initial conditions for Ma et al. (2015 JAMES), while we used ERA-Interim reanalysis as our initial conditions for these multi-year short-range hindcasts. Therefore, these two papers described two different sets of experiments for two different scientific topics. The focus of this present study is on how to better utilize this multi-year hindcasts like the four topics given in the manuscript to address*

*several modeling issues associated with moist processes, which cannot be achieved from previous short Transpose AMIP II hindcasts.*

*Regarding the issue of duplication of work already in the literature, we have never previously published studies on the four topics using multi-year hindcasts described in this study: (1) cloud regimes at the ARM SGP site, (2) model biases associated with MJO, (3) variations of moist processes associated with ENSO, and (4) robustness of systematic errors. It will be helpful if the reviewer can indicate which topic is duplicating existing work in the literature or in our previous studies, and we will revise it or remove the duplicated part entirely.*

**Anonymous Referee #3**

The authors have provided clear responses to most of my comments and made appropriate revisions to the manuscript.

However, I am still concerned about the interpretation of the MJO development and propagation in these hindcasts. In their response, the authors discussed how they formed the 16-year timeseries from hindcasts at a constant lead time (e.g., a 16-year timeseries of day-3 hindcasts). The construction of these timeseries was clear to me in the original manuscript. My concern is how the authors interpret errors in physical processes simulated in the model, based on these timeseries. The authors discuss the pre-conditioning of the active MJO phase by shallow convection, the transition from shallow to deep convection and the propagation of the MJO across the Maritime Continent. These are processes that require more than three days to develop, either in a model or in observations. Thus, I do not believe that the authors can conclude (from these timeseries) that the model fails to simulate these processes correctly, given that the timeseries is discontinuous (i.e., it is made up of one day from each of many hindcasts, stitched together). Any propagation in this timeseries, either for individual events or for the composite, is an artefact of concatenating parts of many different simulations together. The hindcasts do not have sufficient time to simulate the propagation across the Maritime Continent. These processes could be simulated in the nudged simulation, because that is a continuous simulation, which is why I suggested comparing the hindcasts to the nudged run.

The authors need to discuss these caveats so that the reader does not treat the hindcast simulation as a continuous simulation, in which physical processes like pre-conditioning and moistening are simulated within a single integration of the model.

*Response to reviewer:*
*We have added a paragraph to caution the readers about the caveats of using discontinuous time series from the hindcasts concatenating from series of short-range hindcasts in the revised manuscript on Lines 266-270:*

[revised manuscript text omitted]